# Downgrade to Upgrade: Optimizer Simplification Enhances Robustness in LLM Unlearning

**Yicheng Lang**[1]    **Yihua Zhang**[1]    **Chongyu Fan**[1]    **Changsheng Wang**[1]
**Jinghan Jia**[1]    **Sijia Liu**[1,2]
[1]The OPTML Lab, Dept. CSE, Michigan State University
[2]MIT-IBM Watson AI Lab, IBM Research

## Abstract

Large language model (LLM) unlearning aims to surgically remove the influence of undesired data or knowledge from an existing model while preserving its utility on unrelated tasks. This paradigm has shown promise in addressing privacy and safety concerns. However, recent findings reveal that unlearning effects are often *fragile*: post-unlearning manipulations such as weight quantization or fine-tuning can quickly neutralize the intended forgetting. Prior efforts to improve robustness primarily reformulate unlearning objectives by explicitly assuming the role of vulnerability sources. In this work, we take a different perspective by investigating the role of the *optimizer*, independent of unlearning objectives and formulations, in shaping unlearning robustness. We show that the "*grade*" of the optimizer, defined by the level of information it exploits, ranging from zeroth-order (gradient-free) to first-order (gradient-based) to second-order (Hessian-based), is tightly linked to the resilience of unlearning. Surprisingly, we find that downgrading the optimizer, such as using zeroth-order methods or compressed-gradient variants (*e.g.*, gradient sign-based optimizers), often leads to stronger robustness. While these optimizers produce noisier and less precise updates, they encourage convergence to harder-to-disturb basins in the loss landscape, thereby resisting post-training perturbations. By connecting zeroth-order methods with randomized smoothing, we further highlight their natural advantage for robust unlearning. Motivated by these insights, we propose a *hybrid optimizer* that combines first-order and zeroth-order updates, preserving unlearning efficacy while enhancing robustness. Extensive experiments on the MUSE and WMDP benchmarks, across multiple LLM unlearning algorithms, validate that our approach achieves more resilient forgetting without sacrificing unlearning quality. Code is available at https://github.com/OPTML-Group/Unlearn_Optimizer.

## 1 Introduction

Large language models (LLMs) have demonstrated remarkable capabilities in natural language understanding and generation across diverse applications (Achiam et al., 2023; Touvron et al., 2023; Yang et al., 2025a). However, their pre-training on massive data corpora raises growing concerns about safety, privacy, and trustworthiness (Mazeika et al., 2024; Li et al., 2024; Liu et al., 2025; Huang et al., 2024). LLMs may inadvertently reproduce copyrighted content (Eldan & Russinovich, 2023; Shi et al., 2024), expose personally identifiable information (Staab et al., 2023; Yao et al., 2024a), or generate harmful instructions (Barrett et al., 2023; Li et al., 2024). To address these risks, **LLM unlearning** has emerged as a promising direction, aiming to remove the influence of undesired data, knowledge, and associated model capabilities without incurring the cost of retraining the entire model and preserving the model's general utility (Yao et al., 2024b; Fan et al., 2024; Zhang et al., 2024a; Zhuang et al., 2025; Reisizadeh et al., 2025; O'Brien et al., 2025).

Despite recent progress in developing LLM unlearning algorithms that achieve both effective forgetting and utility preservation (Yao et al., 2024b; Zhang et al., 2024a; Fan et al., 2024; Li et al., 2024; Jia et al., 2024a), ensuring **robust** unlearning remains a significant challenge. Unlearning performance can quickly deteriorate under post-unlearning weight perturbations. Prior work shows that fine-tuning on even a small set of forgotten samples or semantically related texts can substantially reverse unlearning effects (Lynch et al., 2024; Hu et al., 2024), while model compression techniques

such as quantization may also resurface erased content (Zhang et al., 2024d). Furthermore, when unlearned models are adapted to downstream tasks via fine-tuning, their unlearning guarantees often degrade (Wang et al., 2025a).

Existing research on robust LLM unlearning has primarily focused on problem-level reformulations or algorithm-level modifications, often assuming a specific vulnerability source and tailoring the unlearning method accordingly. For instance, Fan et al. (2025) cast robust unlearning as a min–max problem against relearning-induced perturbations and adapt sharpness-aware minimization (SAM) (Foret et al., 2020) to strengthen robustness. Tamirisa et al. (2024) propose tamper-resistant unlearning via meta-learning, modeling the attacker as a weight-tampering adversary. Similarly, Wang et al. (2025a) leverage invariant risk minimization (IRM) (Arjovsky et al., 2019) to regularize unlearning against degradation from irrelevant fine-tuning. While effective, these approaches rely on customized changes to unlearning objectives, thereby modifying the underlying optimization algorithm itself. In contrast, the role of the *base optimizer*, independent of any problem-wise and algorithm-level modifications, in shaping unlearning robustness remains largely unexplored. Notably, even heuristic optimizer adjustments, such as increasing the learning rate, have been observed to improve robustness against weight quantization (Zhang et al., 2024d), hinting at a deeper connection. This raises the central research question of this work:

> *(Q) How does the choice of optimizer influence the robustness of LLM unlearning, and what optimizers can improve robustness without sacrificing unlearning effectiveness?*

To address this question, we introduce the concept of *optimizer grade* for LLM unlearning, defined by the level of gradient information utilized by an optimizer. The first-order (FO) gradient-based Adam optimizer (Kingma & Ba, 2014), widely adopted in LLM unlearning (Shi et al., 2024; Li et al., 2024; Jia et al., 2024a; Fan et al., 2024; Zhang et al., 2024d), represents a "high-grade" optimizer. In contrast, *down-graded* alternatives reduce the precision of gradient information. For example, gradient-compression methods such as signSGD and signAdam (Bernstein et al., 2018) quantize gradients into low-bit representations, while zeroth-order (ZO) optimizers rely solely on finite-difference estimates of objective values, serving as gradient-free counterparts to FO methods (Chen et al., 2023; Liu et al., 2020; Zhang et al., 2024c). Although these optimizers reduce gradient fidelity, they remain principled and convergence-guaranteed, making them suitable for solving general optimization tasks, including LLM unlearning.

From the perspective of optimizer grade, a key finding of our work is that *downgrading optimizers can unexpectedly enhance unlearning robustness*. We provide both technical rationale and empirical evidence showing a clear link between optimizer grade and robustness grade. In particular, ZO optimizers, while less precise in unlearning effectiveness, exhibit strong robustness against weight tampering. Building on this insight, we propose a *Hybrid optimizer* that integrates FO and ZO methods within a unified framework, combining the robustness of ZO with the optimization accuracy of FO. In summary, our contributiosn are listed below.

• We present the first systematic study of *optimizer choice* in LLM unlearning, showing that *downgrading* the optimizer (via quantized or zeroth-order updates) can improve robustness against weight tampering. We also provide a rationale: downgraded optimizers introduce higher optimization noise tolerance, making unlearned models more resilient to post-unlearning weight perturbations.

• We propose *FO–ZO hybrid optimization*, a unified framework that integrates FO and ZO optimizers, combining ZO-induced robustness with FO-driven unlearning effectiveness.

• We validate our findings through extensive experiments across diverse unlearning tasks and methods, demonstrating a consistent link between optimizer grade and unlearning robustness.

## 2    RELATED WORKS

**LLM unlearning.** LLM unlearning aims to remove memorized data or specific model behavior from pretrained LLMs (Liu et al., 2025; Fan et al., 2024; Maini et al., 2024; Jia et al., 2024a; Shi et al., 2024). Its applications span copyright protection (Shi et al., 2024; Eldan & Russinovich, 2023), privacy preservation (Wu et al., 2023; Lee et al., 2024; Kuo et al., 2025), and the removal of harmful abilities (Li et al., 2024; Lang et al., 2025; Zhou et al., 2024; Tamirisa et al., 2024)(Wang et al., a). Most existing approaches are fine-tuning based, employing regularized optimization to promote forgetting while retaining general utility (Yao et al., 2024b; Li et al., 2024; Zhang et al., 2024a; Fan et al., 2024; Jia et al., 2024a; Reisizadeh et al., 2025; Yang et al., 2025b)(Wang et al., b;a). Com-

plementary lines of work perform unlearning at inference time without altering model parameters, including in-context unlearning (Thaker et al., 2024; Pawelczyk et al., 2023) and intervention-based decoding strategies (Liu et al., 2024; Suriyakumar et al., 2025; Deng et al., 2025; Bhaila et al., 2025; Wang et al., 2025b).

**Robustness of LLM unlearning.** Recent studies have shown that unlearned LLMs remain vulnerable to both input-level and weight-level "perturbations" (Hu et al., 2024; Lynch et al., 2024; Łucki et al., 2024; Fan et al., 2025). Input-space perturbations, such as in-context examples or adversarial prompts/jailbreaks, can still elicit forgotten information from the model (Łucki et al., 2024; Sinha et al., 2025; Yuan et al., 2025). Weight-space perturbations include quantization, which can resurface memorized data (Zhang et al., 2024d), relearning on forgotten or semantically similar data (Hu et al., 2024; Che et al., 2025; Lynch et al., 2024), and irrelevant downstream fine-tuning that reverses unlearning effects (Wang et al., 2025a). To enhance robustness, several algorithmic defenses have been proposed. Tamper-resistant safeguards leverage meta-learning to anticipate weight tampering (Tamirisa et al., 2024), while latent adversarial training improves resilience in the representation space (Sheshadri et al., 2024). Fan et al. (2025) cast robust unlearning as a min-max optimization problem and apply sharpness-aware minimization (SAM) and smoothness-inducing techniques. Invariant risk minimization (IRM) has been employed to mitigate vulnerabilities from irrelevant fine-tuning (Wang et al., 2025a), and divergence-based regularization, such as Jensen–Shannon divergence, has also been introduced to strengthen robustness (Singh et al., 2025). Beyond optimization strategies, other works explore robust data filtering and pre-training methods to resist harmful weight tampering (O'Brien et al., 2025).

**Optimization for LLM unlearning.** The LLM unlearning problem is typically formulated as an optimization task, making it natural to study through the optimization lens. A notable example is Jia et al. (2024b), who introduced second-order unlearning (SOUL) by linking influence-function-based unlearning (Koh & Liang, 2017) with the second-order optimizer Sophia (Liu et al., 2023), thereby enhancing forgetting performance via iterative influence removal. Similarly, Reisizadeh et al. (2025) leveraged bi-level optimization to balance unlearning effectiveness and utility retention, while Fan et al. (2025) adopted min–max robust optimization to improve resilience. Despite these advances, the role of *optimizer grade* in shaping unlearning robustness has received little attention. In particular, ZO optimization (Liu et al., 2020; Nesterov & Spokoiny, 2017; Duchi et al., 2015; Ghadimi & Lan, 2013), which estimates gradients from function evaluations and finite differences (avoiding backpropagation), has not been studied for LLM unlearning. Initial efforts only applied ZO to non-LLM settings, such as memory-efficient unlearning (Zhang et al., 2025a) and graph unlearning (Xiao et al., 2025), primarily for computational efficiency. Similarly, ZO has also been explored for memory-efficient fine-tuning of LLMs (Malladi et al., 2023; Zhang et al., 2024c; Tan et al., 2025; Mi et al., 2025). In this work, we instead examine ZO from a robust unlearning perspective, showing that, even as a highly degraded form of optimization, it can enhance the resilience of LLM unlearning against weight tampering.

## 3 PRELIMINARIES AND PROBLEM STATEMENT: OPTIMIZER "GRADE" VS. UNLEARNING ROBUSTNESS

**LLM unlearning setup.** LLM unlearning refers to the process of selectively *erasing* the influence of specific data or knowledge (and the associated model behaviors) from a trained model, while preserving its overall usefulness. The aim is to make the model "forget" undesired content (*e.g.*, private, copyrighted, or harmful information) without the cost of retraining from scratch and without impairing its performance on unrelated tasks.

Formally, LLM unlearning is typically cast as a regularized optimization problem involving two competing objectives: a forget loss ($\ell_\mathrm{f}$), which enforces the removal of the undesired data/knowledge, and a retain loss ($\ell_\mathrm{r}$), which preserves the model's general utility. The forget loss is evaluated on the forget dataset $\mathcal{D}_\mathrm{f}$ using an unlearning-specific objective, while the retain loss is computed on the retain dataset $\mathcal{D}_\mathrm{r}$ using standard objectives such as cross-entropy or KL divergence (Maini et al., 2024). This yields the optimization problem (Liu et al., 2025):

$$\underset{\boldsymbol{\theta}}{\text{minimize}} \quad \ell_\mathrm{f}(\boldsymbol{\theta}|\mathcal{D}_\mathrm{f}) + \lambda \ell_\mathrm{r}(\boldsymbol{\theta}|\mathcal{D}_\mathrm{r}), \tag{1}$$

where $\lambda \geq 0$ is a regularization parameter that balances unlearning effectiveness (captured by $\ell_\mathrm{f}$) against utility retention (captured by $\ell_\mathrm{r}$). In (1), the choice of the unlearning objective $\ell_\mathrm{f}$ determines the specific unlearning method applied to solve the problem. For instance, if $\ell_\mathrm{f} = -\ell_\mathrm{r}$, then gradient

descent optimization effectively leverages the gradient difference between prediction losses on $\mathcal{D}_\text{f}$ and $\mathcal{D}_\text{r}$ to promote forgetting. This approach is referred to as Gradient Difference (**GradDiff**) (Liu et al., 2022; Yao et al., 2024b). Alternatively, if $\ell_\text{f}$ is defined via the Direct Preference Optimization (DPO) (Rafailov et al., 2023) objective by treating the forget data in $\mathcal{D}_\text{f}$ exclusively as negative samples, then the resulting negative-sample-only formulation leads to the Negative Preference Optimization (**NPO**) method (Zhang et al., 2024a; Fan et al., 2024) for solving (1). Furthermore, if $\ell_\text{f}$ is cast as a min-max objective against worst-case perturbations (aimed at enhancing unlearning robustness, as will be discussed later), then the resulting forget loss corresponds to the Sharpness-Aware Minimization (**SAM**) objective, giving rise to the SAM-based robust unlearning (Fan et al., 2025).

To *evaluate* unlearning performance, we primarily adopt the **MUSE** benchmark (Shi et al., 2024), which targets copyrighted information removal. MUSE consists of two subsets: unlearning book contents from *Harry Potter* ("books corpus", *MUSE-Books*) and unlearning BBC News articles ("news corpus", *MUSE-News*). Performance is assessed using three metrics: verbatim memorization on the forget set ($\mathcal{D}_\text{f}$; **VerbMem**), knowledge memorization on the forget set ($\mathcal{D}_\text{f}$; **KnowMem**), and knowledge memorization on the retain set ($\mathcal{D}_\text{r}$; **KnowMem**). Unlearning is conducted on two fine-tuned models: ICLM-7B trained on the books corpus and LLaMA2-7B trained on the news corpus. We focus on MUSE because it jointly covers *data-centric* unlearning evaluation (captured by VerbMem) and *knowledge-centric* unlearning evaluation (captured by KnowMem). In addition, we also include experiments on the **WMDP** (Li et al., 2024) and **TOFU** (Maini et al., 2024) benchmarks in the additional experiment section.

**Unlearning robustness challenge.** Once a model has been unlearned to erase undesired information, it is crucial that the forgetting effect remains stable. In other words, the model should be *robust post-unlearning* against both intentional and unintentional weight *perturbations*. In this work, we focus on two representative forms of weight tampering studied in LLM unlearning: *relearning attacks* (Hu et al., 2024; Fan et al., 2025; Deeb & Roger, 2024), which represent *intentional* perturbations aimed at restoring forgotten knowledge, and *weight quantization* (Zhang et al., 2025b), which reflects *unintentional* perturbations introduced by model compression.

*Relearning attacks* exploit data samples that follow the forget data distribution, for example, subsets of $\mathcal{D}_\text{f}$ (Fan et al., 2025; Hu et al., 2024) or retain data $\mathcal{D}_\text{r}$ drawn from the same distribution as $\mathcal{D}_\text{f}$ (Deeb & Roger, 2024). These samples are used to update the unlearned model and test whether the resulting weight perturbations (denoted as $\boldsymbol{\delta}$) can undo the effects of unlearning in (1), thereby resurfacing the forgotten information. Formally, the relearning attack can be expressed as

$$\underset{\boldsymbol{\delta}}{\text{minimize}} \quad \ell_\text{relearn}\left(\boldsymbol{\theta}_\text{u} + \boldsymbol{\delta} \mid \mathcal{D}_\text{relearn}\right), \tag{2}$$

where $\boldsymbol{\theta}_u$ denotes the unlearned model from (1) and $\mathcal{D}_\text{relearn}$ is the relearn dataset. Unless specified otherwise, we set $\mathcal{D}_\text{relearn}$ as a subset of $\mathcal{D}_\text{f}$. Following (Fan et al., 2025), relearn is instantiated by fine-tuning the unlearned model for a fixed number of steps, *e.g.*, 100, which we denote as "*Relearn100*". Different from relearning attacks, *quantization* compresses the full-precision weights of the unlearned model into lower precision by reducing *the number of bits* used to represent them. As shown in (Zhang et al., 2025b), although quantization is a benign compression technique, it can unintentionally undermine unlearning by shifting parameters toward regions in the loss landscape that resurface forgotten knowledge.

**The "grade" of an optimizer: Motivation for its link to unlearning robustness.** Prior work has begun to examine the optimizer's influence on LLM unlearning. Here, we use the term *optimizer* to refer to the objective-agnostic optimization method employed to solve the unlearning problem in (1). For instance, first-order gradient-based methods such as Adam (Kingma & Ba, 2014) can be used to implement multiple unlearning approaches like GradDiff and NPO. It has been shown in (Jia et al., 2024c) that the choice of optimizer can impact unlearning effectiveness. For example, *second-order* optimizers such as *Sophia* (Liu et al., 2023) closely connect to influence function-based unlearning (Koh & Liang, 2017; Jia et al., 2024c), which estimates and removes the effect of specific training data on a model. However, no prior work has examined the optimizer's role in shaping unlearning robustness against weight perturbations like relearning attacks and quantization.

In this work, we introduce a fresh perspective by examining the notion of "*optimizer grade*" and its relationship to the grade of *unlearning robustness*. By "optimizer grade", we refer to the level of (descent) information an optimizer leverages to guide the optimization trajectory converging toward a (locally) optimal solution. We can differentiate the *optimizer grade* based on the *order of gradient information* an optimizer exploits. For instance, zeroth-order (**ZO**) optimization methods (Liu

(a) MUSE-News: Performance on $\mathcal{D}_\mathrm{f}$ and $\mathcal{D}_\mathrm{r}$      (b) MUSE-Books: Performance on $\mathcal{D}_\mathrm{f}$ and $\mathcal{D}_\mathrm{r}$

Figure 1: Unlearning performance under 4-bit weight quantization using NPO on MUSE with different optimizers (Sophia, Adam, 8-bit Adam, and 1-bit Adam). Performance is measured by unlearning effectiveness (VerbMem and KnowMem on $\mathcal{D}_\mathrm{f}$, left plots in each sub-figure) and utility (KnowMem on $\mathcal{D}_\mathrm{r}$, right plots in each sub-figure). "Pre-unlearn" represents the target model to conduct unlearning, and "before Q" (the circle) and "after Q" (the diamond) represent the unlearned models before and after 4-bit weight quantization. (a) Unlearning on MUSE-News. (b) Unlearning on MUSE-Books.

et al., 2020), which approximate gradients through finite differences of objective function values, can be regarded as a *downgrade* of first-order (**FO**) methods; FO methods, in turn, are a *downgrade* of second-order (**SO**) methods. Furthermore, even within the same order, the optimizer grade can vary depending on whether the gradient information is *compressed*. A well-known example is gradient *sign*-based FO optimization, such as signSGD (Bernstein et al., 2018), which represents a *downgrade* of standard SGD. Therefore, we focus on optimizer grades from two perspectives: *(a) inter-order*, comparing zeroth-, first-, and second-order methods; and *(b) intra-order*, contrasting compressed versus uncompressed gradient information within the first order. The **problem of interest** can thus be formulated as: *How does the optimizer grade affect unlearning robustness?*

An interesting and, as we will show later, insightful conclusion is that a *downgraded* optimizer can in fact lead to *upgraded* unlearning robustness. We motivate it by comparing unlearning robustness under 4-bit weight quantization (via GPTQ (Frantar et al., 2022)) across optimizers of varying orders, using NPO on the MUSE benchmark. The optimizers include the SO optimizer *Sophia*, the FO optimizer *Adam*, and its downgraded gradient-compressed variants: *8-bit Adam* (with 8-bit gradient compression) (Dettmers et al., 2022) and *1-bit Adam* (with 1-bit gradient compression, also known as signAdam) (Wang et al., 2019). As shown in **Fig. 1**, before quantization ("before Q"), the unlearning performance of downgraded optimizers (8-bit Adam and 1-bit Adam) is comparable to that of full-precision Adam and Sophia, as indicated by similar VerbMem, KnowMem on $\mathcal{D}_\mathrm{f}$ and KnowMem on $\mathcal{D}_\mathrm{r}$. However, when the unlearned models are subjected to 4-bit quantization for post-unlearning robustness assessment, the unlearning performance of FO Adam and SO Sophia is substantially worse compared to their downgraded optimizer counterparts (*e.g.*, 1-bit Adam), as evidenced by increases in VerbMem and KnowMem on $\mathcal{D}_\mathrm{f}$. By comparison, the SO optimizer Sophia shows the weakest robustness on $\mathcal{D}_\mathrm{f}$ after quantization, even worse than the FO Adam. This highlights a clear interplay between optimizer grade and robustness. Focusing on utility measured by KnowMem on $\mathcal{D}_\mathrm{r}$, we observe that quantized unlearned models gain utility, whereas the original pre-unlearned model suffers a utility drop after quantization. This occurs because quantization can partially revert the unlearning effect, thereby easing the tradeoff between forgetting on $\mathcal{D}_\mathrm{f}$ and retention on $\mathcal{D}_\mathrm{r}$, which in turn boosts utility.

## 4 DOWNGRADING THE OPTIMIZER UPGRADES UNLEARNING ROBUSTNESS

**Optimizer downgrade via gradient compression.** Let $\mathbf{m}_t$ denote the descent direction used in the $t$-th update of a FO optimizer, with the update rule given by $\boldsymbol{\theta}_{t+1} = \boldsymbol{\theta}_t - \eta\mathbf{m}_t$, where $\eta > 0$ denotes a learning rate. For Adam, $\mathbf{m}_t$ corresponds to the momentum term (*i.e.*, moving average of adaptive gradients) (Reddi et al., 2018), while for SGD, $\mathbf{m}_t$ is simply the gradient of the objective function. The gradient compression replaces the full-precision gradient with a quantized version, obtained through a quantization operator $Q(\cdot; N)$ using the gradient's $N$-bit representation:

$$\boldsymbol{\theta}_{t+1} = \boldsymbol{\theta}_t - \eta Q(\mathbf{m}_t; N); \quad \text{And if } N = 1, \text{ then } Q(\mathbf{m}_t; 1) = \mathrm{sign}(\mathbf{m}_t), \qquad (3)$$

where $\mathrm{sign}(\mathbf{x})$ denotes the element-wise sign of the vector $\mathbf{x}$. The SGD variant of (3) with $N = 1$ corresponds to *signSGD* (Bernstein et al., 2018). Similarly, the Adam variants with $N = 8$ and $N = 1$ give rise to *8-bit Adam* (Dettmers et al., 2022) and *signAdam* (Wang et al., 2019), respectively. It is also worth noting that gradient compression reduces the information available in the descent step (3), yet it still suffices to guarantee convergence of the optimization (Bernstein et al., 2018). As shown in Fig. 1, gradient compression improves unlearning robustness compared to its

uncompressed counterpart under post-unlearning weight quantization. This effect can be explained from (3): When a gradient compression-based optimizer is used for unlearning, it *naturally improves tolerance to weight perturbations*, as the quantization operator $Q(\cdot)$ effectively acts as a "denoiser", mapping perturbed weights onto the same discrete bit values.

**Optimizer downgrade via ZO gradient estimation and its link to randomized smoothing.** The observation that gradient compression yields tolerance to weight perturbations suggests a broader principle: if an optimizer inherently tolerates noise, it may also enhance robustness when applied to unlearning. Following this principle, downgrading from FO to ZO optimization can also improve robustness, since ZO methods estimate gradients via finite differences of objective function values, while still enjoying provable convergence guarantees (Liu et al., 2020). Formally, the ZO approximation of the FO gradient $\nabla f(\mathbf{x})$ for an objective function $f(\mathbf{x})$ is given by

$$\widehat{\nabla} f(\mathbf{x}) = \frac{1}{q} \sum_{i=1}^{q} \left[ \frac{f(\mathbf{x} + \mu \mathbf{u}_i) - f(\mathbf{x} - \mu \mathbf{u}_i)}{2\mu} \right] \mathbf{u}_i, \tag{4}$$

where $\{\mathbf{u}_i\}_{i=1}^{q}$ are random direction vectors (*e.g.*, sampled uniformly from the unit sphere), and $\mu > 0$ is the perturbation size used for finite differences. As shown theoretically in (Liu et al., 2018), the ZO gradient estimator is an *unbiased* estimator (4) of the gradient of a *smoothed version* of the original objective function,

$$f_\mu(\mathbf{x}) := \mathbb{E}_{\mathbf{u}}\big[f(\mathbf{x} + \mu \mathbf{u})\big], \quad \text{with} \ \ \nabla f_\mu(\mathbf{x}) = \mathbb{E}_{\mathbf{u}}[\widehat{\nabla} f(\mathbf{x})], \tag{5}$$

where the expectation is taken over the random direction vector $\mathbf{u}$. Therefore, employing a ZO gradient estimation-based optimizer is equivalent to solving a randomized smoothing (**RS**) (Cohen et al., 2019) of the original problem (Liu et al., 2020), where $\widehat{\nabla} f(\mathbf{x})$ serves as a stochastic gradient estimate of the smoothed objective. It is clear from (5) that RS inherently incorporates random noise into the optimization process. Indeed, minimizing an RS-type unlearning objective (with a FO optimizer) has been shown to improve unlearning robustness (Fan et al., 2025).

There exist many variants of ZO optimization methods. For LLM unlearning we emphasize two choices. First, sampling random vectors from the unit sphere distribution rather than a Gaussian yields more stable unlearning by reducing gradient estimation variance (Ma & Huang, 2025). Second, we adopt the **AdaZO** optimizer (Shu et al., 2025), a state-of-the-art method that further reduces variance and improves convergence. Unless otherwise specified, ZO refers to AdaZO.

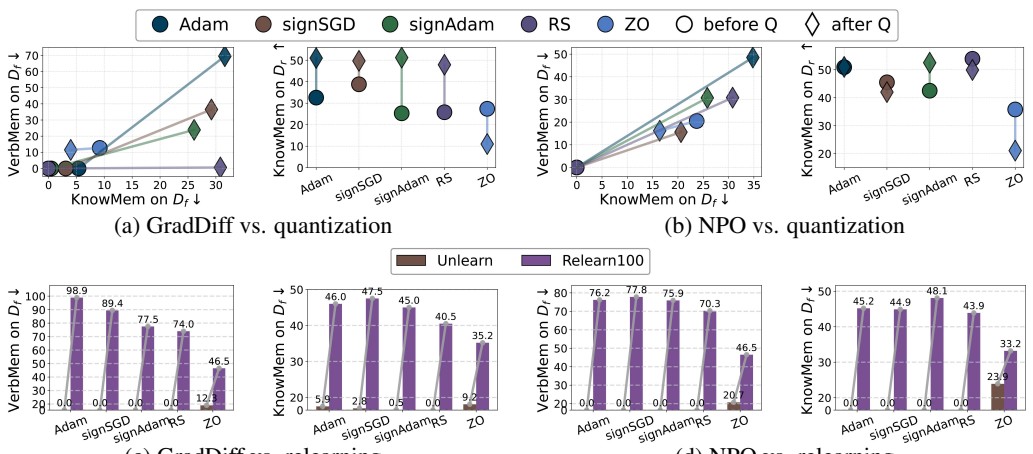

(a) GradDiff vs. quantization      (b) NPO vs. quantization

(c) GradDiff vs. relearning      (d) NPO vs. relearning

Figure 2: On MUSE-Books, (a-b): Unlearning performance under 4-bit weight quantization using GradDiff and NPO with different optimizers (Adam, signSGD, signAdam, (FO) RS, ZO method). The figure format is consistent with Fig. 1. (c-d): Unlearn performance with relearning 100 steps ("Relearn100"), using GradDiff and NPO with different optimizers.

**Enhanced unlearning robustness to weight quantization via downgraded optimizers.** Extending Fig. 1 by incorporating additional downgraded optimizers beyond Adam (including signSGD, RS, and AdaZO), **Fig. 2(a-b)** reports the initial unlearning performance on MUSE-Books ("before Q") and the performance under 4-bit weight quantization ("after Q"), using GradDiff and NPO as the unlearning methods. Consistent with Fig. 1, the 1-bit compressed optimizers signAdam and signSGD improve quantization robustness compared to Adam. Likewise, the first-order RS-based

optimization also achieves both effective unlearning before quantization and improved robustness after quantization. The ZO optimizer, viewed as the ZO downgrade of RS, shows more nuanced behavior. Prior to quantization, ZO exhibits weaker unlearning: for both GradDiff and NPO, it yields higher VerbMem and KnowMem on $\mathcal{D}_f$ and lower KnowMem on $\mathcal{D}_r$. However, after quantization, ZO demonstrates *remarkably strong robustness*: it attains substantially lower VerbMem and KnowMem on $\mathcal{D}_f$ than other methods. This pattern holds across both GradDiff- and NPO-based unlearning. The tradeoff is that ZO yields the weakest utility, reflecting its downgraded optimization accuracy. As will be shown later, we can leverage ZO's robustness benefits to improve FO-based unlearning via a hybrid approach that integrate ZO with FO.

**ZO exhibits stronger robustness than other optimizers against relearning.** Fig. 2(c-d) shows unlearning robustness under *relearning attacks*. Among first-order downgraded optimizers, FO RS performs the best, with lower VerbMem and KnowMem on $\mathcal{D}_r$ after 100 relearning steps ("Relearn100") for both GradDiff and NPO, consistent with literature on smoothness optimization (Fan et al., 2025). In contrast, signAdam and signSGD show only occasional gains over Adam. The most notable improvement comes from ZO, which consistently yields the lowest VerbMem and KnowMem on $\mathcal{D}_f$ across both unlearning methods. Results on robustness to relearning (Fig. 2(c-d)) and weight quantization (Fig. 2(a-b)) highlight the distinctive advantage of the downgraded ZO optimizer over RS, gradient-compressed FO, and standard FO. We hypothesize that ZO guides unlearning into a different optimization basin, yielding distinct dynamics and greater robustness.

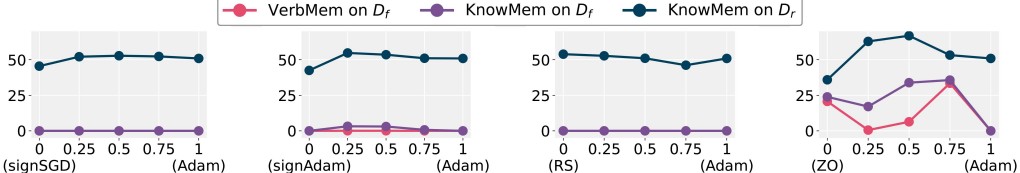

Figure 3: Linear mode connectivity (LMC) between downgraded optimizers (signSGD, signAdam, RS, and ZO) and Adam on MUSE-Books, using NPO.

To validate the distinctiveness of ZO optimizers, we use **linear mode connectivity (LMC)** (Frankle et al., 2020; Qin et al., 2022; Lubana et al., 2023; Pal et al., 2025) to compare converged unlearning solutions from two optimizers. LMC assesses whether two unlearned models can be connected by linear interpolation in parameter space. Formally, for $\boldsymbol{\theta}_1$ and $\boldsymbol{\theta}_2$, LMC holds if the unlearning metric (*e.g.*, KnowMem on $\mathcal{D}_f$) of $\boldsymbol{\theta}(\alpha) = \alpha\boldsymbol{\theta}_1 + (1-\alpha)\boldsymbol{\theta}_2$ remains consistent as $\alpha \in [0,1]$ varies. **Fig. 3** shows LMC between models unlearned with downgraded optimizers and Adam. Gradient-compressed optimizers (signSGD, signAdam) display clear connectivity with Adam: VerbMem on $\mathcal{D}_f$, KnowMem on $\mathcal{D}_f$, and KnowMem on $\mathcal{D}_r$ remain stable across interpolation, indicating convergence to the same basin. In contrast, ZO lacks LMC with Adam, implying convergence to a *separate basin* supporting its distinctive unlearning and robustness.

## 5   BEST OF BOTH WORLDS: LLM UNLEARNING VIA HYBRID OPTIMIZATION

As indicated by Fig. 3, FO and ZO optimizers converge to different basins: FO yields stronger unlearning but limited robustness, whereas ZO offers weaker unlearning before quantization and relearning yet greater robustness to weight perturbations, due to the perturbation tolerance of its gradient estimation and optimization. This raises the question of *whether integrating ZO into FO can achieve both effective unlearning and robustness beyond the standard FO optimizer*.

Recall that ZO is inherently noisier than FO due to gradient estimation variance (4), which limits optimization efficiency (Liu et al., 2020). To address this, we propose a **hybrid FO–ZO method** ("Hybrid"): FO optimization (Adam by default) is applied to the pre-unlearned model $\boldsymbol{\theta}$ for $N$ steps, producing $\boldsymbol{\theta}_N$; then ZO optimization (AdaZO by default) continues for another $N$ steps to obtain $\boldsymbol{\theta}_{2N}$. This alternation repeats, ending on a ZO round, so the final model is $\boldsymbol{\theta}_{kN}$ for $k$ alternating rounds.

**Rationale behind FO-ZO hybrid: A follower-leader game.** In the proposed hybrid strategy, the alternation between FO and ZO naturally integrates their optimization effects. This can be viewed as a *two-player game*: the high-grade FO optimizer acts as a player that solves the unlearning problem with high precision, while the ZO optimizer introduces noise, effectively solving a random-smoothing objective that enhances tolerance to weight perturbations. However, we find that starting

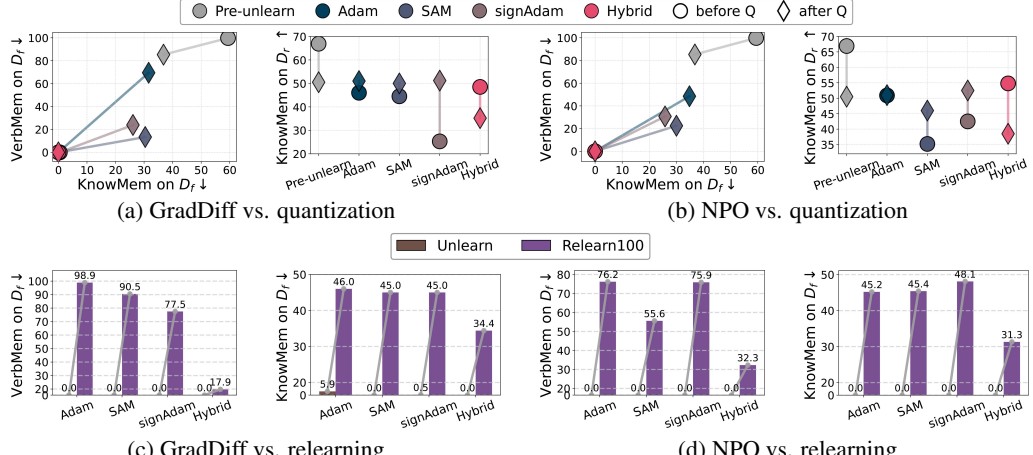

Figure 4: (a–b): Unlearning performance before and after 4-bit quantization on MUSE-Books using GradDiff and NPO with optimizers Adam, SAM, signAdam, and Hybrid FO–ZO. (c–d): GradDiff and NPO on MUSE-Books under different optimizers against "Relearn100" (100 relearning steps). The figure format follows Fig. 2.

with the FO optimizer and ending with the ZO optimizer yields stronger unlearning robustness and a more stable optimization process, consistent with our design goal. The rationale behind the hybrid schedule is that this two-player game can be viewed as a *leader–follower game (also known as bi-level optimization)* (Zhang et al., 2024b). Since unlearning robustness is the primary goal, the ZO optimizer should be treated as the "leader." Meanwhile, the FO optimizer, as a high-grade optimizer with stronger unlearning effectiveness, acts as the "follower," providing a high-quality initialization for ZO and reducing the variance introduced by ZO gradient estimation.

**Hybrid optimization achieves both strong unlearning effectiveness and robustness.** In **Fig. 4**, we show that the "Hybrid" optimizer demonstrates superior robustness to both weight quantization (Fig. 4(a–b)) and relearning (Fig. 4(c–d)), outperforming gradient-compressed signAdam, standard Adam, and SAM (sharpness-aware minimization with explicit robust design). As shown in Fig. 4(a–b), before quantization, Hybrid achieves superior unlearning effectiveness, evidenced by the lowest VerbMem and KnowMem scores on $\mathcal{D}_{\mathrm{f}}$, while preserving utility as measured by KnowMem on $\mathcal{D}_{\mathrm{r}}$. This stands in sharp contrast to the ZO optimizer in Fig. 2, where robustness gains come at the cost of utility loss. After quantization, Hybrid maintains consistent robustness benefits, with utility drops similar to those of the original model. Notably, its robustness gains even surpass SAM, despite SAM's explicit robustness design in the unlearning objective. Fig. 4(c–d) further demonstrates Hybrid's robustness against relearning. For both GradDiff- and NPO-based unlearning, Hybrid achieves substantially lower VerbMem and KnowMem after Relearn100.

## 6 ADDITIONAL EXPERIMENTS

In this section, we provide additional experiments validating the link between optimizer grade and unlearning robustness grade, including evaluations on WMDP (Li et al., 2024), TOFU (Maini et al., 2024), and supportive experiments for our proposal.

**Experiment setups.** We further evaluate on the **WMDP** benchmark, which tests harmful knowledge removal via LLM unlearning. Following the robustness protocol in (Fan et al., 2025), we fine-tune unlearned models on a small subset of forget samples for varying epochs. Experiments use `Zephyr-7B-beta` with two stateful unlearning algorithms: representation misdirection for unlearning (RMU) (Li et al., 2024) and NPO. Baselines include Adam, signAdam, ZO, and SAM, compared against our Hybrid. Unlearning effectiveness is measured by test accuracy on WMDP-Bio, while utility is measured by accuracy on MMLU (Hendrycks et al., 2020); effective unlearning corresponds to low WMDP-Bio and high MMLU accuracy. We further validate the proposed Hybrid method on the **TOFU** benchmark (Maini et al., 2024), designed for fictitious unlearning on a synthetic QA dataset. Using NPO under the *forget10* scenario, the goal is to erase memorization of fictitious authors. The target model is `LLaMA2-7B` fine-tuned on the dataset corpus. Evaluation uses three metrics: (i) Probability on $\mathcal{D}_{\mathrm{f}}$ (Prob.), (ii) ROUGE-L on $\mathcal{D}_{\mathrm{f}}$ (Rouge), and (iii) Model utility (MU), aggregating memorization on $\mathcal{D}_{\mathrm{r}}$, real authors, and world knowledge. Effective unlearning corresponds to low Prob./Rouge and high MU.

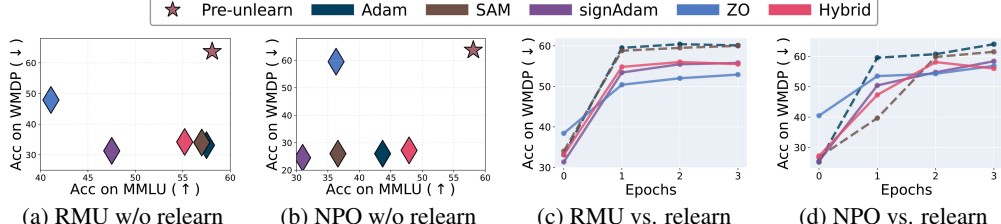

(a) RMU w/o relearn     (b) NPO w/o relearn     (c) RMU vs. relearn     (d) NPO vs. relearn

Figure 5: Unlearning performance and relearning robustness of RMU and NPO on WMDP-Bio using different optimizers (Adam, signAdam, ZO, SAM, and Hybrid). Relearning is conducted by fine-tuning the unlearned model on 40 forget data samples across multiple epochs. (a) Unlearning effectiveness and utility retention of RMU without relearning; (b) NPO without relearning; (c) RMU across different relearning epochs; (d) NPO across different relearning epochs.

**Experiment results on WMDP.** As WMDP unlearning is vulnerable to relearning attacks Fan et al. (2025), we investigate the role of optimizers before and after such attacks. Relearning is simulated by fine-tuning the unlearned model on 40 forget samples across epochs. **Fig. 5** shows WMDP and MMLU accuracy for RMU and NPO (**a-b**), and robustness under relearning (**c-d**). The proposed Hybrid consistently outperforms baselines in both settings, notably surpassing SAM—despite its explicit robustness design—while retaining comparable or superior unlearning effectiveness before relearning. Another notable observation is that when robustness against relearning is not considered, the ZO optimizer appears inferior to other methods in Fig. 5(a-b), owing to its high optimization variance from ZO gradient estimation, consistent with the MUSE results in Fig. 2. However, once relearning is taken into account, the robustness benefit of ZO becomes evident in Fig. 5(c-d), even surpassing Hybrid at larger relearning epochs. This again confirms our key finding that *downgrading the optimizer can enhance the robustness of unlearning*.

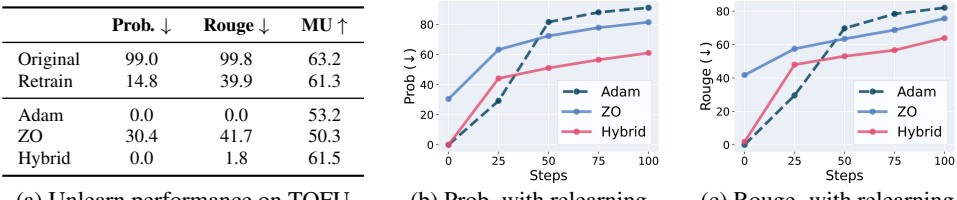

| | Prob. ↓ | Rouge ↓ | MU ↑ |
|---|---|---|---|
| Original | 99.0 | 99.8 | 63.2 |
| Retrain | 14.8 | 39.9 | 61.3 |
| Adam | 0.0 | 0.0 | 53.2 |
| ZO | 30.4 | 41.7 | 50.3 |
| Hybrid | 0.0 | 1.8 | 61.5 |

(a) Unlearn performance on TOFU     (b) Prob. with relearning.     (c) Rouge. with relearning

Figure 6: Unlearning performance and robustness of NPO using Adam, ZO, and Hybrid optimizer on TOFU under the forget10 scenario. (a) Unlearning effectiveness of NPO before relearning with different optimizers, evaluated by probability (Prob.), ROUGE-L (Rouge), and model utility (MU). Here, "Original" denotes the pre-unlearned target model, while "Retrain" refers to the model trained solely on the retain dataset, provided by TOFU. (b–c) Robustness against relearning, showing Prob. and Rouge. against increasing relearning steps.

**Experiments on TOFU. Fig. 6** presents the NPO-based unlearning performance on TOFU before and after relearning using different optimizers (Adam, ZO, and Hybrid). As shown in Fig. 6(a), Hybrid consistently matches or outperforms Adam, achieving stronger unlearning effectiveness with lower Prob. and Rouge. and higher MU. In contrast, ZO delivers weaker unlearning prior to relearning. However, Fig. 6(b–c) highlights the robustness advantage of ZO and Hybrid over Adam under relearning, as both maintain lower Prob. and Rouge values with increasing steps. Notably, Hybrid provides the best overall trade-off, combining effective unlearning with resilience to relearning, outperforming Adam and enjoying ZO's robustness.

**Ablation studies on hybrid optimization.** We conduct additional experiments on MUSE-Books to provide further justification for the optimizer scheduler design in Hybrid optization, detailed in **Sec. 5**. As shown in **Fig. 7**, changing the switch step $N$ does not materially affect unlearning's robustness against quantization and relearning. Besides, **Fig. 8** presents the performance where Hybrid optimization has different steps $N$ for FO and ZO. We can see that allocating an equal number of FO and ZO updates (denoted as FO= 20, ZO= 20) achieves the best balance between unlearning effectiveness and robustness. This outcome is consistent with our method design grounded in the leader–follower game (**Sec. 5**). Assigning more FO than ZO steps (*e.g.*, FO= 20, ZO= 10) results in a decline in unlearning robustness because the "leader" component (ZO updates responsible for steering the model toward robustness) becomes weaker than the "follower" (FO updates that emphasize high-precision unlearning but do not explicitly promote robustness). Conversely, assigning

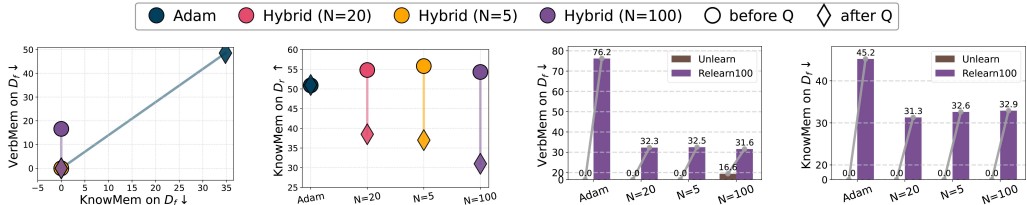

(a) NPO vs. quantization MUSE-Books      (b) NPO vs. relearning MUSE-Books

Figure 7: (a) NPO-based unlearn performance and quantization robustness of Hybrid optimization with switch steps 20, 5 and 100 (*e.g.*, "Hybrid ($N = 20$)" represents Hybrid optimization where the FO and ZO optimizer switch every 20 steps). (b) Relearning robustness of Hybrid optimization with different switch steps.

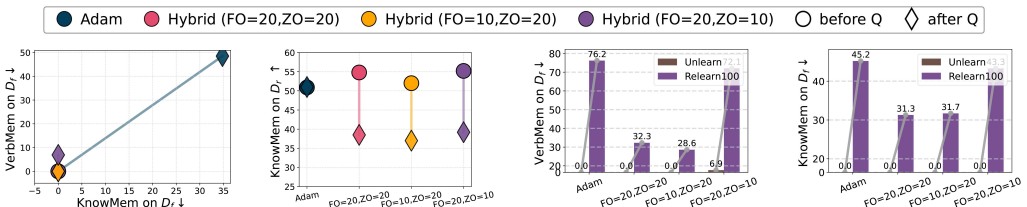

(a) NPO vs. quantization on MUSE-Books      (b) NPO vs. relearning on MUSE-Books

Figure 8: (a) NPO-based unlearn performance and quantization robustness of Hybrid optimization where FO and ZO have different steps (*e.g.*, "FO= 10, ZO= 20" represents optimization with Adam for 10 steps and ZO for 20 steps in each round), evaluated on MUSE-Books. (b) Relearning robustness of Hybrid optimization with different FO and ZO steps.

more ZO than FO steps (*e.g.*, FO= 10, ZO= 20) slightly reduces unlearning effectiveness, since the "follower" (FO) becomes weaker to provide the high-fidelity updates needed to maintain strong unlearning performance in a non-relearning evaluation setting.

**Other ablation studies.** In Appx. E, we further validate the robustness of Hybrid by conducting relearning experiments on both $\mathcal{D}_f$ and $\mathcal{D}_r$ for different numbers of steps, and additionally include *general utility* (*i.e.*, model capabilities that should be preserved but are not explicitly tested in unlearning benchmarks) evaluation for the optimizers discussed in this study. As detailed in the appendix, Hybrid demonstrates consistent robustness on both $\mathcal{D}_f$ and $\mathcal{D}_r$, and lower-grade optimizers do not necessarily compromise general utility. Additionally, our Hybrid method is highly efficient and does not incur additional runtime cost.

## 7 CONCLUSION

To enhance the robustness of LLM unlearning against post-unlearning weight tampering (*e.g.*, relearning attacks and weight quantization), we investigate the role of optimizer design and demonstrate that downgrading the optimizer can improve robustness. This reveals a novel connection between optimizer grade and unlearning robustness. Among downgraded optimizers, zeroth-order (ZO) methods show weaker unlearning performance (when weight tampering is not considered) but substantially greater robustness compared to first-order (FO) optimizers for unlearning. Building on this insight, we propose a FO-ZO hybrid optimization strategy that augments standard FO unlearning with ZO updates, achieving both strong unlearning effectiveness and enhanced robustness. Extensive experiments across multiple datasets validate the benefits of this approach. We refer readers to Appx. A–C for discussions on limitations, ethics statement, and LLM usage.

## 8 ACKNOWLEDGEMENT

This work was supported in part by the National Science Foundation (NSF) CISE Core Program Awards IIS-2207052 and IIS-2504263, the NSF CAREER Award IIS-2338068, the Cisco Research Award, the Amazon Research Award for AI in Information Security, the Open Philanthropy Research Award, the Schmidt Sciences Research Award, and the Center for AI Safety (CAIS) Compute Award.

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

APPENDIX

## A  LIMITATIONS

While we conduct comprehensive experiments and in-depth analysis to show the role of optimizers in robust LLM unlearning, certain limitations persist in our study. There are other optimizers we did not include in our study, *e.g.*, the Muon optimizer and the Shampoo optimizer. Also, our methods and insights could be extended to relevant and important fields, such as safety alignment, which we did not include in this work. Additionally, there needsstudy on whether the downgrade of optimizers improves robustness in general.

## B  ETHICS STATEMENT

The datasets used in this paper are from publicly available sources and do not contain sensitive or private information. Our research focuses on the LLM unlearning, which erases private or harmful data memorization in LLMs and enhances LLM safety. By studying optimizer design and integrating hybrid optimization, we further improve the robustness of unlearning, making it less vulnerable to post-unlearning weight tampering.

## C  LLM STATEMENT

In this paper, the sole purpose of LLMs is to assist with improving the fluency of the paper, such as refining the grammar. At no point did the language model contribute to research ideas or to the generation of original content.

## D  DETAILED EXPERIMENT SETUP

**Settings on MUSE.**  For the first-order (FO) optimizers (Adam, gradient-compressed Adam, and RS) on both MUSE-Books and MUSE-News, we fix $\beta$ in NPO to $0.1$, and tune the learning rate in the range $[5e-6, 1e-5]$. On MUSE-Books, we perform unlearning for 1 epoch and tune the retain loss coefficient $\lambda$ for GradDiff and NPO in $\{1.0, 10.0, 20.0, 50.0\}$ via grid search. On MUSE-News, we conduct 5 epochs of unlearning, saving checkpoints per epoch, and select the checkpoint with the best retain performance as the final model.

For the zeroth-order (ZO) methods and Hybrid, we also fix $\beta$ in NPO to $0.1$ and make the same grid search for $\lambda$. We tune the learning rate via grid search in $[1e-5, 5e-5]$ and conduct 1000 steps of unlearning, checkpointing every 100 steps to select the model with the best retain performance. In Hybrid, we switch the optimizer every 20 steps on MUSE-Books and every 50 steps on MUSE-News.

**Settings on WMDP.**  For both NPO and RMU using FO optimizers, we perform 150 unlearning steps. For NPO, we fix $\beta$ to $0.1$ and tune the hyperparameters via grid search: learning rate $\in$ $[5e-6, 1e-5]$ and $\lambda \in \{1.0, 2.5\}$. For RMU, we follow the default settings proposed in Li et al. (2024).

For NPO and RMU using ZO and Hybrid, we perform 400 unlearning steps and checkpoint every 100 steps, selecting the model with the best utility. We tune the learning rate via grid search in $[1e-5, 5e-5]$ and employ the same $\lambda$ values as in the FO setting. For Hybrid, we switch the optimizer every 20 steps.

**Settings on TOFU.**  We fix $\beta$ in NPO to $0.1$ and tune $\lambda \in \{1.0, 2.5\}$. For the FO setting, we fix the learning rate to $1e-5$. For ZO and Hybrid, we tune the learning rate via grid search in $[1e-5, 5e-5]$. For Hybrid, we switch the optimizer every 20 unlearning steps.

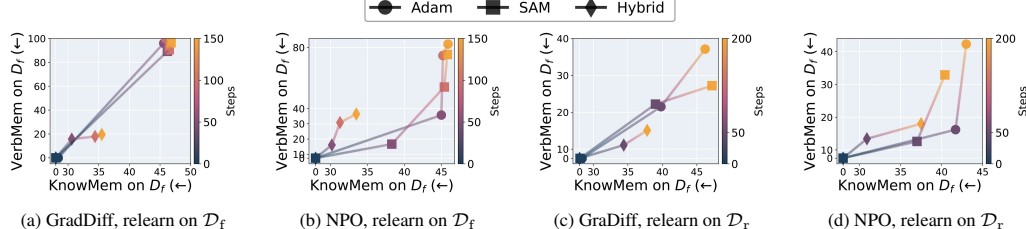

Figure A1: Robustness of GradDiff and NPO on MUSE-Books against relearning on $\mathcal{D}_f$ and $\mathcal{D}_r$ across different numbers of relearning steps. The initial unlearned models at "step 0" are obtained using Adam, SAM, and Hybrid optimizers, respectively.

# E  ABLATION STUDIES.

**Additional experiments on hybrid optimization.**  We evaluate the robustness of the proposed Hybrid optimizer under two relearning settings: using the forget set $\mathcal{D}_f$ and the retain set $\mathcal{D}_r$. While earlier experiments considered the worst-case robustness scenario with $\mathcal{D}_f$ as relearning samples, our results show that Hybrid maintains robustness even when the relearning set is drawn from $\mathcal{D}_r$, demonstrating its resilience beyond the worst-case setting. **Fig. A1** shows that Hybrid consistently outperforms Adam and SAM, achieving lower KnowMem and VerbMem on $\mathcal{D}_f$ across the relearning path. Moreover, Hybrid not only surpasses SAM with its explicit robustness design against relearning attacks but also demonstrates stable resilience when fine-tuned on $\mathcal{D}_r$.

**General utility evaluation.**  We employ the following benchmarks and the `lm-eval-harness` library (Gao et al., 2024) to evaluate the general utility of the unlearned models. These benchmarks target different aspects of reasoning, factuality, and commonsense competence:

- **Hellaswag** (Zellers et al., 2019) measures a model's ability to perform commonsense reasoning in everyday situations. It presents a context and several possible sentence completions. High performance indicates strong capability in narrative understanding and choosing contextually appropriate continuations.
- **TruthfulQA** (Lin et al., 2021) evaluates the truthfulness and factual of model responses.
- **ARC-Challenge** (Clark et al., 2018) focuses on scientific question answering at the level of elementary and middle school multiple-choice exams.

As shown in Table. A1, models trained with down-graded optimizers and Hybrid etain competitive performance relative to the upgraded FO optimizer, confirming that lower-grade optimizers do not necessarily compromise general utility.

Table A1: General utility evaluation of different optimizers across unlearning benchmarks and methods. The values in the table are the *average* of hellaswag, truthfulqa and arc evaluation scores.

| Dataset | Pre-unlearn | Method | Optimization Methods | | | | |
| --- | --- | --- | --- | --- | --- | --- | --- |
| | | | Adam | signAdam | SAM | ZO | Hybrid |
| MUSE-Books | 36.2 | GradDiff | 32.6 | 32.9 | 28.1 | 26.4 | 30.1 |
| | | NPO | 27.2 | 28.5 | 27.6 | 32.9 | 25.2 |
| MUSE-News | 41.9 | GradDiff | 37.0 | 37.7 | 37.4 | 36.5 | 36.7 |
| | | NPO | 38.8 | 40.9 | 32.7 | 37.0 | 37.3 |
| WMDP | 53.3 | RMU | 53.4 | 49.1 | 52.9 | 41.5 | 48.9 |
| | | NPO | 33.1 | 30.0 | 26.7 | 41.1 | 35.9 |

**Run time evaluation.**  As shown in **Table. A2**, downgraded optimizers such as ZO and Hybrid achieve comparable wall-clock time to standard first-order optimizers. While ZO uses multiple function evaluations, the absence of backward passes and the limited number of unlearning iterations make its cost practically similar. We also observe that downgraded optimizers are significantly *more efficient* than the robust-optimization baseline SAM, whose sharpness-aware updates require

Table A2: Run time (in minutes) of different optimizers for GradDiff and NPO-based unlearning on MUSE.

| Dataset | Unlearning Objective | Optimization Method | | | | | | |
|---------|---------------------|------|--------|----------|------|------|------|--------|
| | | Adam | signSGD | signAdam | RS | SAM | ZO | Hybrid |
| MUSE-Books | GradDiff | 15.2 | 14.8 | 15.0 | 15.4 | 30.6 | 18.9 | 17.7 |
| | NPO | 18.9 | 17.7 | 17.8 | 18.1 | 35.9 | 22.8 | 21.9 |
| MUSE-News | GradDiff | 33.2 | 31.1 | 32.6 | 35.9 | 85.7 | 40.8 | 41.7 |
| | NPO | 39.3 | 38.5 | 39.0 | 39.8 | 81.3 | 40.9 | 39.3 |

an expensive inner loop. In contrast, our Hybrid method alternates between FO and ZO updates without introducing any nested optimization, preserving efficiency while improving robustness.

## F ADDITIONAL RESULTS

We show the unlearning performance with quantization and relearning for GradDiff and NPO on *MUSE-News*, using the down-graded optimizers, in **Fig. A2**. LMC of NPO on MUSE-News is shown in **Fig. A3**. We further show the performance of Hybrid on MUSE-News with NPO and GradDiff, against relearning and quantization, in **Fig. A4**.

For both the study of downgraded optimizers and hybrid optimization, the experiment results on MUSE-News are aligned with MUSE-Books: For instance, as shown in Fig. A2(a-b), ZO achieves the best performance with 4-bit quantization ("with Q"). Fig. A2(c-d) further demonstrates the robustness of ZO against relearning, where ZO with both GradDiff and NPO acheives the lowest KnowMem and VerbMem on $\mathcal{D}_\mathrm{f}$ after relearning 100 steps.

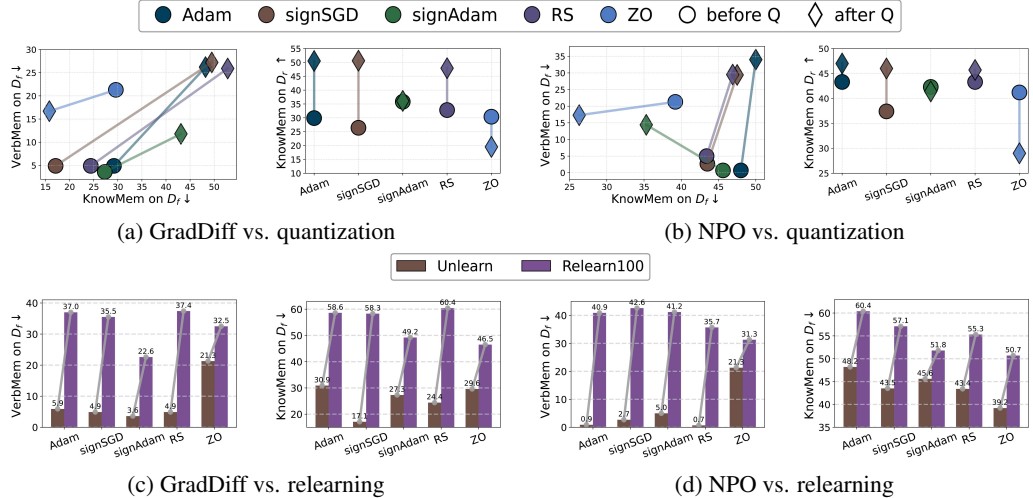

Figure A2: Unlearning performance and robustness using GradDiff and NPO on MUSE-News with different optimizers (Adam, signSGD, signAdam, (FO) RS, ZO method). (a-b) shows unlearning's robustness against 4-bit quantization, and (c-d) shows unlearning's robustness against relearning 100 steps ("Relearn100").

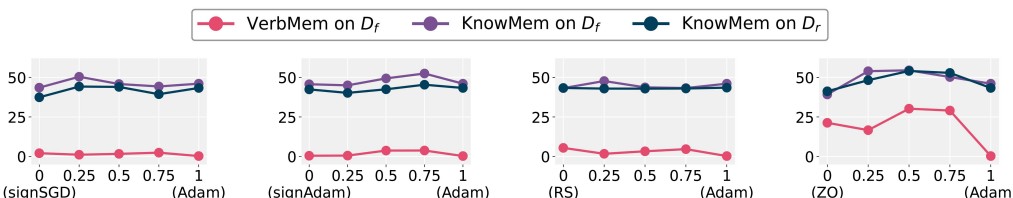

Figure A3: Linear mode connectivity (LMC) between downgraded optimizers (signSGD, signAdam, RS, and ZO) and Adam on MUSE-News, using NPO for unlearning.

The effectiveness of hybrid optimization is also demonstrated on MUSE-News, as Fig. A4 illustrates. Across GradDiff and NPO, Hybrid yields unlearn performance on par with Adam. Especially with the NPO algorithm, Hybrid shows a clear robustness advantage against both quantization (Fig. A4(b)) and relearning (Fig. A4(d)) compared to the baseline optimizers (*e.g.*, Adam and SAM).

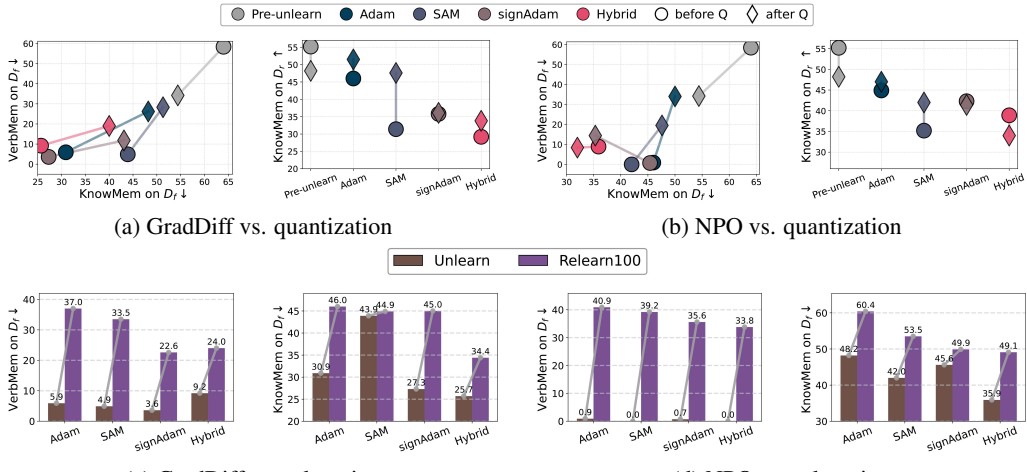

(a) GradDiff vs. quantization          (b) NPO vs. quantization

(c) GradDiff vs. relearning          (d) NPO vs. relearning

Figure A4: (a-b):Unlearning performance before and after 4-bit quantization using GradDiff and NPO on MUSE-News with the optimization methods: Adam, sharpness-aware minimization (SAM), signAdam and hybrid FO-ZO optimization (Hybrid). (c-d): GradDiff and NPO with different optimizers against relearning 100 steps. The figure format is consistent with Fig. 2.

| Method | VerbMem (↓) | | KnowMem (↓) | | Retain (↑) | | Utility (↑) | | |
|---|---|---|---|---|---|---|---|---|---|
| | W/o Q | W/ Q | W/o atk | W/ atk | W/o atk | W/ atk | Truthful-QA | Hellaswag | ARC-Challenge |
| Pre-unlearn | 99.8 | 85.3 | 59.4 | 36.8 | 66.9 | 50.5 | 21.4 | 50.0 | 37.3 |
| NPO | 0 | 48.5 | 0 | 34.8 | 53.65 | 51.0 | 23.3 | 31.0 | 27.3 |
| NPO w/ signSGD | 0 | 15.6 | 0 | 20.6 | 44.5 | 42.0 | 22.2 | 35.8 | 31.7 |
| NPO w/ signAdam | 0 | 30.7 | 0 | 25.8 | 35.9 | 52.5 | 23.6 | 33.2 | 28.8 |
| NPO w/ RS | 0.0 | 34.5 | 0 | 23.4 | 54.6 | 49.9 | 23.4 | 31.1 | 28.2 |
| NPO w/ SAM | 0.0 | 30.0 | 0.0 | 22.5 | 35.2 | 46.0 | 23.6 | 29.7 | 26.7 |
| NPO w/ ZO | 20.7 | 16.2 | 23.9 | 16.4 | 36.6 | 21.1 | 18.5 | 44.7 | 35.5 |
| NPO w/ Hybrid | 0 | 0 | 0 | 0 | 54.8 | 38.5 | 23.8 | 28.4 | 23.5 |
| GradDiff | 0 | 60.5 | 5.9 | 31.6 | 46.0 | 51.0 | 22.4 | 41.7 | 33.6 |
| GradDiff w/ signSGD | 0 | 36.5 | 2.81 | 29.2 | 36.7 | 49.7 | 21.9 | 42.2 | 33.0 |
| GradDiff w/ signAdam | 0 | 23.9 | 0.5 | 26.1 | 25.3 | 51.2 | 20.8 | 42.7 | 35.3 |
| GradDiff w/ RS | 0 | 0.61 | 0 | 27.35 | 26.8 | 47.9 | 22.2 | 39.4 | 34.9 |
| GradDiff w/ SAM | 0 | 13.4 | 0 | 30.4 | 44.5 | 50.0 | 21.4 | 42.1 | 33.5 |
| GradDiff w/ ZO | 12.3 | 11.5 | 9.2 | 4.0 | 26.8 | 11.0 | 20.3 | 36.7 | 27.4 |
| GradDiff w/ Hybrid | 0 | 0 | 0 | 0 | 48.5 | 38.2 | 24.1 | 30.8 | 24.2 |

Table A3: Unlearning evaluation and general utilities on MUSE-Books, using GradDiff and NPO.

| Method | VerbMem (↓) | | KnowMem (↓) | | Retain (↑) | | Utility (↑) | | |
|---|---|---|---|---|---|---|---|---|---|
| | W/o Q | W/ Q | W/o atk | W/ atk | W/o atk | W/ atk | Truthful-QA | Hellaswag | ARC-Challenge |
| Pre-unlearn | 58.4 | 34.2 | 64.0 | 54.4 | 55.2 | 48.2 | 26.9 | 56.2 | 42.7 |
| NPO | 0.9 | 34.0 | 48.2 | 50.0 | 43.4 | 47.0 | 26.6 | 52.4 | 37.5 |
| NPO w/ signSGD | 2.7 | 29.4 | 43.5 | 47.5 | 37.4 | 46.0 | 26.4 | 51.9 | 37.9 |
| NPO w/ signAdam | 0.7 | 14.4 | 45.6 | 35.3 | 42.3 | 41.4 | 28.9 | 53.9 | 40.0 |
| NPO w/ RS | 5.0 | 29.5 | 43.4 | 46.9 | 43.3 | 45.7 | 26.8 | 53.0 | 37.0 |
| NPO w/ SAM | 0.0 | 19.5 | 42.0 | 47.6 | 35.2 | 42.0 | 26.1 | 42.7 | 29.4 |
| NPO w/ ZO | 21.3 | 17.3 | 39.2 | 26.3 | 41.2 | 29.0 | 23.5 | 50.7 | 36.8 |
| NPO w/ Hybrid | 8.9 | 8.4 | 35.9 | 32.0 | 38.9 | 34.0 | 26.4 | 49.7 | 35.7 |
| GradDiff | 5.9 | 26.2 | 30.9 | 48.2 | 46 | 51.5 | 26.9 | 56.2 | 42.7 |
| GradDiff w/ signSGD | 4.9 | 27.2 | 17.1 | 49.5 | 26.4 | 50.6 | 26.1 | 49.0 | 37.2 |
| GradDiff w/ signAdam | 3.6 | 11.8 | 27.3 | 43.1 | 35.8 | 36.1 | 25.2 | 49.3 | 36.4 |
| GradDiff w/ RS | 4.9 | 25.9 | 24.4 | 52.8 | 32.8 | 47.9 | 26.1 | 49.4 | 37.6 |
| GradDiff w/ SAM | 4.9 | 28.3 | 43.9 | 51.3 | 31.4 | 47.6 | 26.8 | 46.2 | 37.0 |
| GradDiff w/ ZO | 21.3 | 16.7 | 29.6 | 15.8 | 30.4 | 19.5 | 2.6 | 50.3 | 36.6 |
| GradDiff w/ Hybrid | 9.2 | 19.0 | 25.7 | 40.0 | 29.2 | 33.8 | 23.3 | 52.2 | 35.9 |

Table A4: Unlearning evaluation and general utilities on MUSE-News, using GradDiff and NPO.

