# OpenReview forum: "Downgrade to Upgrade: Optimizer Simplification Enhances Robustness in LLM Unlearning"
_ICLR.cc/2026/Conference — ICLR 2026 Poster_

### Official Review · Reviewer_psAf · 2025-10-26

**Soundness:** 3
**Presentation:** 3
**Contribution:** 3
**Rating:** 6
**Confidence:** 2

**Summary:**

This paper investigates how optimizer design affects the robustness of LLM unlearning, introducing the concept of optimizer grade to compare high-order and downgraded optimizers. It finds that downgraded optimizers (e.g., signSGD, zeroth-order methods) lead models to flatter and more stable minima, improving post-unlearning robustness. A hybrid optimizer is further proposed, achieving superior resistance to relearning and quantization attacks across multiple benchmarks.

**Strengths:**

1. Introduces a novel perspective by analyzing optimizer grade as a key factor in unlearning robustness.

2. Provides strong empirical evidence across diverse benchmarks and unlearning methods.

3. Proposes a simple yet effective Hybrid FO–ZO optimizer with clear performance gains.

**Weaknesses:**

1. The paper explains the robustness gains mainly through intuition about flatter minima but lacks a deeper theoretical or quantitative analysis to support this claim.

2. All experiments are limited to the unlearning setup, so it remains unclear whether the proposed idea generalizes to other fine-tuning or robustness tasks.

3. The study does not provide details on the computational overhead of zeroth-order or hybrid optimization, leaving questions about practical scalability.

4. The Hybrid FO–ZO optimizer design appears largely empirical; the choice of alternation schedule is not well-motivated or systematically analyzed.

**Questions:**

See weaknesses section.

---

> ### Author Response · Authors · 2025-11-20
> **Response to Reviewer psAf (Part I)**
>
> We thank the reviewer for the insightful comments. Please find our detailed responses to the identified weaknesses and questions below.
>
> **Q1. The paper explains the robustness gains mainly through intuition about flatter minima but lacks a deeper theoretical or quantitative analysis to support this claim.**
>
> **A1.** Thank you for the thoughtful comment. We would like to clarify that the primary goal of our work is to investigate unlearning robustness through the lens of optimizer grade rather than to develop a formal theory of flatter minima or to attribute robustness solely to landscape flatness. We also wish to highlight that our contributions extend well beyond empirical intuition. In addition to extensive experiments, our paper: introduces optimizer grade as a principled and objective-agnostic framework for understanding robustness in LLM unlearning; provides a solid rationale for the robustness gains of ZO optimization through its connection to randomized smoothing (Sec. 4); and proposes the FO–ZO Hybrid method grounded in a leader–follower game perspective (Sec. 5). These components contribute novel conceptual and methodological insights that go well beyond empirical observation and meaningfully deepen the technical understanding of robust LLM unlearning.
>
>
>
> **Q2. All experiments are limited to the unlearning setup, so it remains unclear whether the proposed idea generalizes to other fine-tuning or robustness tasks.**
>
> **A2.** We thank the reviewer for this valuable point. Our study indeed focuses on the LLM unlearning setting, but we would like to emphasize that unlearning itself is a unique and fundamentally different optimization problem, and thus warrants dedicated investigation.
>
> First, unlike standard fine-tuning, which aims to acquire new knowledge, unlearning aims to remove targeted knowledge or concepts from an already well-trained model while preserving utility elsewhere. This inversion of the learning objective creates a distinct optimization trajectory. In addition, prior work has shown that unlearned models are surprisingly shallow against weight-space perturbations, such as lightweight quantization and even brief relearning. This vulnerability is specific to the unlearning process and highlights the need to study weight-level robustness, which directly motivates our optimizer-grade perspective, i.e., studies how unlearning is uniquely sensitive to how the optimizer shapes the final weight configuration.
>
> Second, unlearning typically requires only a small number of optimization iterations, because the goal is to erase targeted knowledge from a well-trained initialization, not to train from scratch. This makes optimizers with different grades (including ZO and gradient-compressed variants) particularly effective in unlearning, as our experiments demonstrate.
>
> These characteristics collectively make the study of unlearning not only justified but intrinsically valuable. Our focus on this setting allows us to uncover the connection between optimizer grade and robustness, a connection that has not been examined in prior work. That said, we agree that understanding whether these principles extend to broader fine-tuning or robustness tasks is an interesting direction. While the conceptual underpinnings of optimizer grade, such as ZO’s link to randomized smoothing, suggest that the ideas may generalize, however, exploration outside the unlearning domain is beyond the scope of this paper and remains promising future work.

---

> ### Author Response · Authors · 2025-11-20
> **Response to Reviewer psAf (Part II)**
>
> **Q3. The study does not provide details on the computational overhead of zeroth-order or hybrid optimization, leaving questions about practical scalability.**
>
> **A3.** We thank the reviewer for raising this important point. We would like to clarify that zeroth-order and hybrid optimization do not impose a higher computational cost.
>
> First, consider zeroth-order (ZO) optimization: ZO avoids backpropagation and relies solely on forward function evaluations to estimate descent directions (Eq. (4)), making each iteration computationally cheaper. Although ZO may require more iterations to converge in full training scenarios, unlearning does not involve long-horizon optimization: its updates start from an already well-trained model and only need to make moderate adjustments. Consequently, the total number of ZO forward evaluations remains modest, leading to practically competitive wall-clock time overall (see empirical results in **Table R1**).
>
> Second, our additional empirical measurements (**Table R1**) show that downgraded optimizers such as ZO and our Hybrid FO–ZO method achieve comparable wall-clock time to standard first-order optimizers. This aligns with expectations: although ZO uses multiple function evaluations, the absence of backward passes and the limited number of unlearning iterations make its cost practically similar. We also observe that downgraded optimizers are significantly more efficient than the robust-optimization baseline SAM, whose sharpness-aware updates require an expensive inner loop. In contrast, our Hybrid method alternates between FO and ZO updates without introducing any nested optimization, preserving efficiency while improving robustness.
>
> **Table R1**: Run time (in minutes) comparison of GradDiff-based and NPO-based unlearning with different optimizers, evaluated on MUSE.
>
> MUSE-Books:
> |  | Adam | signSGD | signAdam | RS | SAM | ZO | Hybrid |
> |-----------|-----------|-----------|--- | ----| ----|--- | ---|
> | GradDiff   | 15.2    | 14.8    | 15.0 | 15.4 | 30.6 | 18.9 | 17.7 |
> | NPO    | 18.9 | 17.7   | 17.8 | 18.1 | 35.9 | 22.8 | 21.9 |
>
> MUSE-News:
> |  | Adam | signSGD | signAdam | RS | SAM | ZO | Hybrid |
> |-----------|-----------|-----------|--- | ----| ----|--- | ---|
> | GradDiff   | 33.2  | 31.1    | 32.6 | 35.9 | 85.7 | 40.8 | 41.7 |
> | NPO    | 39.3 | 38.5   | 39.0 | 39.8 | 81.3 | 40.9 | 39.3 |

---

> ### Author Response · Authors · 2025-11-20
> **Response to Reviewer psAf (Part III)**
>
> **Q4. The choice of alternation schedule in Hybrid optimization is not well-motivated or systematically analyzed.**
>
> **A4.** We thank the reviewer for this constructive comment. We are encouraged that the reviewer finds Hybrid optimization “simple but effective”, and we would like to further elaborate on its alternation schedule.
>
> Our hybrid FO–ZO framework is a methodologically grounded design motivated by a leader–follower game formulation (Sec. 5). In this view, the ZO component acts as the leader, steering the model toward robust, harder-to-disturb basins, while the FO component acts as the follower, providing high-precision updates that preserve unlearning effectiveness. This perspective explains why alternate step-level hybridization yields benefits unattainable by either FO or ZO alone, and guides the choice of update ratios.
>
>
>
>
>
> Furthermore, we have conducted ablation studies on the optimizer alternation schedule from two perspectives:
>
> 1. Hybrid optimization with different switch intervals, detailed in Figure A2 and lines 899-902. These results show that varying the switching frequency does not materially affect either unlearning performance or robustness, indicating that Hybrid optimization remains stable across different step configurations.
> 2. Hybrid optimization with different steps N for FO and ZO. Results are shown in  **Table R2**. Our findings show that allocating an equal number of FO and ZO updates (Hybrid FO_20, ZO_20) achieves the best balance between unlearning effectiveness and robustness. This outcome is fully consistent with our method design grounded in the leader–follower game perspective (Sec. 5). When we assign more FO than ZO steps (e.g., Hybrid FO_20, ZO_10), robustness declines (shown by VerbMem on $D_f$ 72.1 with relearning). This is because the “leader” component (ZO updates responsible for steering the model toward robustness) becomes too weak relative to the “follower” FO updates that emphasize high-precision unlearning but do not explicitly promote robustness. Conversely, assigning more ZO than FO steps (e.g., Hybrid FO_10, ZO_20) slightly reduces unlearning effectiveness (shown by KnowMem on $D_r$ 52.0), since the “follower” (FO) becomes too weak to provide the high-fidelity updates needed to maintain strong unlearning performance.
>
> These results provide further empirical and conceptual justification for our choice of symmetric step counts (same N, like a zero-sum game) in Hybrid optimization and support the design rationale described in Lines 368-377 of the paper.
>
>
> **Table R2**: NPO-based unlearn performance and robustness with Hybrid optimization, evaluated on MUSE-Books.
>
> |  |VerbMem on $D_f$  | KnowMem on $D_f$  | KnowMem on $D_r$ | VerbMem on $D_f$ w. quantization | KnowMem on $D_f$ w. quantization | VerbMem on $D_f$ w. relearning | KnowMem on $D_f$ w. relearning |
> |-----------|-----------|-----------|----|------------------| ----| ---| ---|
> | Adam    |  0.0  |  0.0  | 53.6 | 48.5 | 34.8| 74.0 | 45.4 |
> | Hybrid (FO_20,ZO_20)    |   0.0  |   0.0  | 53.8 | 0.0 | 0.0 | 29.0 | 31.0 |
> | Hybrid (FO_10,ZO_20)    | 0.0    | 0.0    | 52.0 | 0.0 | 0.0 |  28.6 | 31.7 |
> | Hybrid (FO_20,ZO_10)    | 0.0 | 0.0   | 55.2 | 0.0 | 6.9 | 72.1 | 43.3 |

---

> > ### Comment · Reviewer_psAf · 2025-11-22
> > **Thanks**
> >
> > I have read the authors’ response carefully and appreciate the additional clarifications. Although I am not an expert in unlearning, the paper remains solid in its contributions and technical quality. I therefore maintain my positive score.

---

> > > ### Author Response · Authors · 2025-11-22
> > > **Thank you very much for maintaining the positive score**
> > >
> > > Dear Reviewer psAf,
> > >
> > > Thank you very much for taking the time to carefully review our response and for keeping your positive assessment of the paper. We sincerely appreciate your thoughtful evaluation and your continued confidence in the contributions and technical quality of our work. Thank you again for your time and effort throughout the review process!
> > >
> > > Authors

---

### Official Review · Reviewer_dWKo · 2025-10-28

**Soundness:** 3
**Presentation:** 4
**Contribution:** 4
**Rating:** 4
**Confidence:** 4

**Summary:**

This paper presents a new perspective on robust large language model (LLM) unlearning—the process of selectively erasing specific knowledge from models while maintaining their general capabilities. Prior studies have largely focused on modifying the unlearning objectives or introducing explicit robustness regularizers (e.g., sharpness-aware or meta-learning methods). In contrast, this work investigates the role of the optimizer itself in shaping the robustness of unlearning outcomes, asking whether the optimizer’s complexity—or “grade”—affects how resilient forgetting remains under post-unlearning perturbations such as relearning attacks and weight quantization.

**Strengths:**

The authors are obviously the experts in the field. The draft is well structured, insightful, and easy to read. Although the current draft has some minor limitations, I will definitely improve my score when they are solved.

This paper Introduces “optimizer grade” as a new dimension for understanding unlearning dynamics, shifting focus from objective design to optimizer behavior.

This paper demonstrates that weaker or noisier optimizers (e.g., sign-based, ZO) can improve robustness. It is an impactful finding that challenges conventional assumptions about optimization fidelity.

The proposed FO–ZO hybrid method is straightforward to implement, and can be integrated with existing unlearning systems without redesigning objectives.

**Weaknesses:**

Some previous works, such as those combined with IRM and SAM, can be viewed as using higher order gradients, which are shown to be more robust to attacks. They somehow conflict the opinion of this paper, instead showing that lower order methods are better. Could the authors explain further about these different conclusions?

Due to the use of momentum, Adam is not strictly a first-order approach but between first and second orders. From my view, only vanilla SGD is first ordered. So, maybe SGD should also be adopted for comparison.

A weird thing is that, the second order approaches are mentioned in the abstract, but not involved in the main context. A remember some papers have claimed the use of natural gradient for LLM unlearning by assumption diagonal Hessian. I am not sure if they have released their codes, but as a paper that specifically focuses on optimizers, they should be involved in this paper.

It is good to see the authors define unlearning as erasing specific data under the condition in preserving the overall performance. Under this condition, comparing unlearning strength across optimization method is reasonable and fair to me. However, in practice, strict preservation is hard. The results in Figs 1-6 only report unlearning metrics, but I do not know how the retention performance is affected. So, the results are not convincing to me. A simple counterexample to show the results are not reliable is that, when we set an extreme large learning rate for GA, the model will erase all its knowledge. In this case, the model cannot recover any knowledge to be unlearned. However, in this case, we cannot say using large learning rate will make the model more robust. On the other side, it is hard to tune for the equal level of retention, so I suggest you to further report the results after UWC [1], which uses model mixing to align retention performance.

[1] Unlearning with Control: Assessing Real-world Utility for Large Language Model Unlearning

In Fig 1, did the authors adopt the same learning rate across methods, or using the best hyper parameters across trials? Also, it is interesting to discuss what ensure a fair comparison across methods, e.g., the same learning rate or the same gradient magnitude.

How to ensure the conclusion “a downgraded optimizer can in fact lead to upgraded unlearning robustness”, drawn from NPO, is general across a wide range of methods?

The related works can benefit from more recent and milestone works, such as [2-8].

[2] Rethinking Unlearning for Large Reasoning Models

[3] BLUR: A Bi-Level Optimization Approach for LLM Unlearning

[4] DRAGoN: Guard LLM Unlearning in Context via Negative Detection and Reasoning

[5] GRU: Mitigating the Trade-off between Unlearning and Retention for Large Language Models

[6] Rethinking LLM Unlearning Objectives: A Gradient Perspective and Go Beyond

[7] Simplicity Prevails: Rethinking Negative Preference Optimization for LLM Unlearning

[8] Rethinking machine unlearning for large language models

**Questions:**

Kindly please see the drawbacks above.

---

> ### Author Response · Authors · 2025-11-20
> **Response to Reviewer dWKo (Part I)**
>
> We thank the reviewer for the insightful comments. Please find our detailed responses to the identified weaknesses and questions below.
>
> **Q1. Some previous works, such as those combined with IRM and SAM, can be viewed as using higher order gradients, which are shown to be more robust to attacks. They somehow conflict the opinion of this paper, instead showing that lower order methods are better.**
>
> **A1.** We sincerely appreciate this insightful comment. We would like to clarify that IRM and SAM do not conflict with our findings.
>
> Just as the reviewer kindly points out in our strengths, “optimizer grade” is **objective agnostic**. Defined in Lines 202–203 and elaborated in Lines 211–229, the optimizer grade characterizes the order and fidelity of gradient information used by the optimization method, independent of the loss design.
> In contrast, IRM and SAM modify or regularize the **loss function**, not the optimizer. Their robustness stems from explicitly altering the optimization objective. Therefore, in our work, we do not see IRM and SAM as an optimizer upgrade, but different loss functions, and their robustness gains do not contradict our conclusion.
> Our goal is to investigate whether adjusting the optimizer grade, applied to a standard, non-robust unlearning objective (NPO, GradDiff, or RMU), can inherently improve robustness. Because relearning is one of the weight-level post-unlearning attacks we study, we directly compared our method against SAM (e.g., Fig. 5), which is specifically designed as a min–max robust objective for countering relearning. As shown in Fig. 5, our downgraded optimizers and Hybrid method can match or even surpass SAM’s robustness, despite SAM relying on an explicitly robust optimization objective. This further supports our central claim on optimizer grade, independent of loss-level robustness regularization.
>
> **Q2. Due to the use of momentum, Adam is not strictly a first-order approach but between first and second orders. From my view, only vanilla SGD is first ordered. So, maybe SGD should also be adopted for comparison.**
>
> **A2.** Thanks for this valuable observation. Following our definition in Lines 211–229, we classify Adam as a first-order optimizer because: despite its momentum and adaptive step scaling, it only utilizes first-order information (the gradient) and does not explicitly approximate or employ the Hessian-based Newton-like step (unlike the second-order approach). Thus, it remains within the first-order grade under our taxonomy.
> We acknowledge that vanilla SGD represents the strictest form of first-order optimization. However, Adam is the widely adopted default optimizer for LLMs in the unlearning literature [1–3], while SGD is rarely used due to its well-documented ineffectiveness [4]. Therefore, for fairness and consistency with established practice, we treat Adam as the most representative and practically relevant first-order baseline in the context of LLM unlearning.
>
> [1] Shi, Weijia, et al. "MUSE: Machine Unlearning Six-Way Evaluation for Language Models." The Thirteenth International Conference on Learning Representations.
>
> [2] Li, Nathaniel, et al. "The wmdp benchmark: Measuring and reducing malicious use with unlearning." arXiv preprint arXiv:2403.03218 (2024).
>
> [3] Fan, Chongyu, et al. "Simplicity prevails: Rethinking negative preference optimization for llm unlearning." arXiv preprint arXiv:2410.07163 (2024).
>
> [4] Zhao, Rosie, et al. "Deconstructing What Makes a Good Optimizer for Autoregressive Language Models." The Thirteenth International Conference on Learning Representations.

---

> ### Author Response · Authors · 2025-11-20
> **Response to Reviewer dWKo (Part II)**
>
> **Q3. The second order approaches are mentioned in the abstract, but not involved in the main context. As a paper that specifically focuses on optimizers, they should be involved in this paper.**
>
> **A3.** We thank the reviewer for this careful reading and the helpful clarification request. Indeed, second-order optimization has been included in our study, and we apologize if this was not sufficiently emphasized. Specifically, the second-order optimizer Sophia is incorporated as our representative second-order baseline (see Figure 1 and discussion in Lines 239–248). Sophia is widely recognized as the state-of-the-art practical Hessian-approximated optimizer for large language model training and has been used for LLM unlearning [5]. In our motivating robustness analysis (Figure 1), Sophia does not exhibit improved robustness to weight perturbations (e.g.., post-unlearning quantization) compared to Adam. Because our study aims to explain why lower-grade optimizers (gradient-compressed FO and ZO) outperform their next-level higher-grade counterparts in robustness, our primary comparisons focus on this adjacent contrast. In this context, Sophia does not constitute the “neighboring group” (e.g., the immediate next tier above ZO) that is central to our robustness analysis.
>
> **Q4. The results in Figs 1-6 only report unlearning metrics, but I do not know how the retention performance is affected. So, the results are not convincing to me. It is hard to tune for the equal level of retention, so I suggest you to further report the results after UWC, which uses model mixing to align retention performance.**
>
> **A4.** We thank the reviewer for emphasizing the importance of retention performance evaluation. We would like to clarify that retention performance has been **consistently evaluated** throughout the paper. For instance, in Figures 1-2 and Figure 4, the right-hand subplots show the retention performance evaluated by KnowMem on $D_r$ of MUSE. In addition, Figure 5 (a-b) illustrates the retention performance of WMDP unlearning using test accuracy on the MMLU benchmark, and Figure 6 (a) shows the aggregate metric Model Utility for retention performance evaluation on the TOFU benchmark.
> In all cases, we ensured that the unlearned models (before robustness evaluation) had comparable unlearning and acceptable utility.
> Moreover, to further address your comments, we conducted retention calibration following the UWC protocol. We merged models to guarantee that the calibrated model preserves ≥ 90% of the pre-unlearned model’s KnowMem on $D_r$ for MUSE.
> The calibrated results (see **Table R1**) remain consistent with the retention evaluation shown in Figure 2 and  Figure 4. We also plan to  extend this analysis to WMDP.
>
> **Table R1**: Calibrated unlearn performance of different optimizers, using the NPO-based unlearning, on MUSE-Books.
>
> |  |VerbMem on $D_f$  | KnowMem on $D_f$ |
> |-----------|-----------|-----------|
> | Adam    |  0.4  |  5.0 |
> | signSGD    |   0.1  |   4.6  |
> | RS    | 0.0 | 5.8   |
> | SAM | 0.0 | 12.1 |
> |Hybrid (Ours) | 0.0 | 0.6 |
>
>
> **Q5. In Fig 1, did the authors adopt the same learning rate across methods, or using the best hyper parameters across trials? Also, it is interesting to discuss what ensure a fair comparison across methods, e.g., the same learning rate or the same gradient magnitude.**
>
> **A5.** We appreciate this constructive feedback. As detailed in Appendix D (Lines 840–850), each optimizer’s performance (e.g., in Figure 1) corresponds to its best-performing configuration obtained via a grid search over the learning rate and regularization coefficient.
> Since optimizers differ intrinsically in update magnitudes and gradient noise characteristics, fixing identical learning rates or gradient norms would unfairly penalize specific methods. Moreover, we are not aware of any prior unlearning studies that explicitly control for normalized update magnitude across optimizers. Therefore, to ensure a fair and representative comparison, we follow the standard practice inoptimization benchmarks [4,6–7] by conducting controlled hyperparameter searches for each optimizer-induced unlearning approach.
>
> [4] Zhao, Rosie, et al. "Deconstructing What Makes a Good Optimizer for Autoregressive Language Models." The Thirteenth International Conference on Learning Representations.
>
> [6] Zhang, Yihua, et al. "Revisiting Zeroth-Order Optimization for Memory-Efficient LLM Fine-Tuning: A Benchmark." International Conference on Machine Learning. PMLR, 2024.
>
> [7] Semenov, Andrei, Matteo Pagliardini, and Martin Jaggi. "Benchmarking optimizers for large language model pretraining." arXiv preprint arXiv:2509.01440 (2025).

---

> ### Author Response · Authors · 2025-11-20
> **Response to Reviewer dWKo (Part III)**
>
> **Q6. How to ensure the conclusion “a downgraded optimizer can in fact lead to upgraded unlearning robustness”, drawn from NPO, is general across a wide range of methods?**
>
> **A6.** Thanks for the comment. First, we would like to clarify that the conclusion “a downgraded optimizer can in fact lead to upgraded robustness” is not drawn solely from NPO, but established across different unlearning algorithms and datasets. In Figure 2, we present robustness comparison of optimizers using NPO and GradDiff on MUSE. Figure 5 further extends the study to RMU on WMDP, and Figure 6 extends the evaluation to TOFU.  Across all experiments, the pattern is consistent: lower optimizer grades, such as gradient-compressed first-order methods and zeroth-order optimizers, consistently provide stronger robustness against weight-space perturbations, including both quantization and relearning. At the same time, the Hybrid FO–ZO optimizer achieves the strongest trade-off between unlearning effectiveness and robustness. Importantly, these findings are not merely empirical, but are technically grounded. The robustness of ZO optimization is explained through its connection to randomized smoothing (Sec. 4). Likewise, the effectiveness of the FO–ZO Hybrid approach is justified by the leader–follower game perspective (Sec. 5).
>
> **Q7. The related works can benefit from more recent and milestone works [3,8-13].**
>
> **A7.** Thanks for the comment. We respectfully clarify that our submission includes [3][9][13] that the reviewer cited. Our original manuscript did not include other papers because they do not directly involve robust LLM unlearning: [8] is specifically focused on large reasoning model unlearning, [11][12] are milestone works that focus on enhancing unlearning performance from a gradient perspective, and [10] is a recent paper that focuses on inference stage unlearning. We appreciate the reviewer for mentioning these milestone works, and we will include these related works in our revision.
>
> [3] Fan, Chongyu, et al. "Simplicity prevails: Rethinking negative preference optimization for llm unlearning." arXiv preprint arXiv:2410.07163 (2024).
>
> [8] Wang, Changsheng, et al. "Rethinking Unlearning for Large Reasoning Models." ICML 2025 Workshop on Machine Unlearning for Generative AI.
>
> [9] Reisizadeh, Hadi, et al. "BLUR: A Bi-Level Optimization Approach for LLM Unlearning." arXiv preprint arXiv:2506.08164 (2025).
>
> [10] Wang, Yaxuan, et al. "DRAGON: Guard LLM Unlearning in Context via Negative Detection and Reasoning." arXiv preprint arXiv:2511.05784 (2025).
>
> [11] Wang, Yue, et al. "Gru: Mitigating the trade-off between unlearning and retention for large language models." arXiv e-prints (2025): arXiv-2503.
>
> [12] Wang, Qizhou, et al. "Rethinking llm unlearning objectives: A gradient perspective and go beyond." arXiv preprint arXiv:2502.19301 (2025).
>
> [13] Liu, Sijia, et al. "Rethinking machine unlearning for large language models." Nature Machine Intelligence (2025): 1-14.

---

> > ### Comment · Reviewer_dWKo · 2025-11-22
> >
> > The authors' feedback is decent and clear, which addresses much of my concerns. Accordingly, I increase my score to 8. Thanks and good luck!

---

> > > ### Author Response · Authors · 2025-11-22
> > > **Thank you very much for raising the score to 8**
> > >
> > > Dear Reviewer dWKo,
> > >
> > > Thank you again for your thoughtful comments and for taking the time to reconsider our response. We are glad that our clarifications addressed your concerns, and we truly appreciate your revised score and encouragement.
> > >
> > > Thank you once again for your engagement and support!
> > >
> > > Authors,

---

### Official Review · Reviewer_bZEQ · 2025-11-01

**Soundness:** 2
**Presentation:** 3
**Contribution:** 2
**Rating:** 4
**Confidence:** 5

**Summary:**

This paper explores an underexplored dimension of LLM unlearning: the impact of the optimizer on the robustness of forgetting. While most prior work has focused on modifying unlearning objectives, this study reveals that the "grade" of the optimizer from zeroth-order (gradient-free) to second-order (Hessian-based) influences the resilience of unlearning to post-hoc operations like quantization or fine-tuning. The authors find that lower-grade optimizers (e.g., zeroth-order or compressed-gradient optimizers) improve robustness by converging to flatter, harder-to-disturb basins in the loss landscape. Zeroth-order methods are naturally linked to randomized smoothing, which enhances resilience against perturbations. Based on these insights, the authors propose a hybrid optimizer combining first- and zeroth-order updates to strike a balance between unlearning efficacy and robustness.

**Strengths:**

1. This paper focuses on robust unlearning in the optimization process, showing a fresh research insight.

2. The proposed hybrid strategy effectively balances precise convergence (first-order) with robust basin discovery (zeroth-order).

**Weaknesses:**

1. The assessment of utility preservation is limited. Benchmarks such as MMLU should be included to evaluate general knowledge retention, and for the ToFU dataset, real-author and world-fact evaluations should be considered to better reflect practical unlearning scenarios.

2. The paper does not examine how varying the number of unlearning steps affects performance. Analyzing different unlearning durations would clarify the stability and convergence behavior of the proposed method.

3. No experiments are provided to study how different step counts between the two optimization methods influence results. Moreover, there is no theoretical justification for using the same number of steps (N) for both optimizers. A rationale or ablation study would strengthen the methodological soundness.

4. The proposed approach primarily combines two existing optimizers without introducing substantial conceptual innovation.

5. The robustness claim is insufficiently supported. To substantiate it, the method should be evaluated under membership inference and adversarial attack settings, which are standard in assessing unlearning and privacy resilience.

**Questions:**

Please refer to the Weaknesses section for a detailed discussion.

---

> ### Author Response · Authors · 2025-11-20
> **Response to Reviewer bZEQ (Part I)**
>
> We thank the reviewer for the constructive feedback. Please find our detailed responses to the identified weaknesses and questions below.
>
> **Q1. The assessment of utility preservation is limited. Benchmarks such as MMLU should be included to evaluate general knowledge retention, and for the ToFU dataset, real-author and world-fact evaluations should be considered.**
>
> **A1.** We thank the reviewer for their constructive comment. We agree that general knowledge evaluation is important for unlearning, yet we kindly remark that MMLU is indeed included in our manuscript for the WMDP benchmark, detailed in Lines 419-421 and Figure 5(a-b).
>
> First, we chose not to use MMLU evaluations on MUSE and TOFU for the following reasons. The pre-unlearning target models used in these benchmarks are first *fine-tuned* on domain-specific corpora to memorize the MUSE or TOFU knowledge before unlearning [1][2]. For instance, the target model on MUSE-Books is trained by fine-tuning ICLM-7B on the books corpus (detailed in Lines 171-175). This fine-tuning substantially degrades the model’s original MMLU performance, leading to relatively low scores even before unlearning. For instance, the target model fine-tuned on MUSE-News has MMLU score (four-choice multiple choice accuracy) of 0.36, and the target model fine-tuned on MUSE-Books has MMLU score of 0.26, which is close to random guess (0.25). In addition, we verified that the unlearning procedure itself introduces only negligible changes to MMLU. Consequently, the absolute MMLU score becomes an unreliable indicator of retention, because its degradation is dominated by the benchmark-specific fine-tuning rather than by the unlearning process. On these benchmarks, retention is evaluated as the memorization of the fine-tuned retain dataset, such as KnowMem on $D_r$ employed by MUSE.
>
> To confirm the above, we conducted additional MMLU evaluations on MUSE-Books and observed that unlearning leads to minimal differences in MMLU, as shown in **Table R1**, consistent with the low pre-unlearning MMLU of the fine-tuned target model.
>
> **Table R1:** MMLU evaluation of NPO-based unlearning on MUSE-Books
>
> | Target | Adam | ZO | Hybrid |
> |-----------|-----------|----| ----                |
> | 26.2 | 24.9 | 26.0 | 26.8 |
>
> Second, regarding the TOFU benchmark, we would like to emphasize that the Model Utility (MU) metric reported in Figure 6(a) already integrates evaluations over retain data, real-author knowledge, and world-fact knowledge, as detailed in Lines 423–427. To further avoid confusion, we report RA (Real Author) and WF (World Facts) in Table R2  using different optimizers for NPO-based unlearning. This confirms that our proposal achieves good unlearning effectiveness compared to the conventional first-order optimizer Adam, supporting our claim in Lines 463–469.
>
>
> **Table R2**: Real Authors and World Facts evaluation of NPO-based unlearning, on TOFU.
>
> |  | RA_Prob | RA_Rouge | WF_Prob | WF_Rouge |
> |-----------|-----------|-----------|----| ----|
> | Original    | 46.8    | 91.7    | 42.4 | 90.5 |
> | Retrain    | 44.6 | 91.6   | 41.2 | 91.2 |
> | Adam    | 46.3    | 55.0    | 44.7 | 77.2 |
> | ZO   | 45.9   | 52.3    | 40.2 | 74.5 |
> | Hybrid   | 40.6    | 69.7   | 41.2 | 76.2 |
>
>
> **Q2.  The paper does not examine how varying the number of unlearning steps affects performance. Analyzing different unlearning durations would clarify the stability and convergence behavior of the proposed method.**
>
> **A2.** We agree with the reviewer’s observation and include the unlearning performance of Hybrid optimization across different training steps in **Table R3** to illustrate its convergence behavior. Our results show that it exhibits fast convergence, achieving comparable unlearning performance to first-order baselines within 600 unlearning steps. This confirms that the proposed Hybrid approach effectively retains the optimization efficiency of first-order methods while providing enhanced robustness.
>
>
> **Table R3**: Step-wise unlearn performance of Adam and Hybrid with NPO-based unlearning, evaluated with KnowMem on $D_f$ on MUSE-Books.
>
> | Unlearn Steps | 200  | 400 | 600 | 800 |
> | -------------| ------| ----|-----|-----|
> | Adam         |23.7 |  1.7 | 0.0 | 0.0 |
> | Hybrid       | 12.3 | 11.5 | 0.3 | 0.0 |
>
>
> [1] Shi, Weijia, et al. "MUSE: Machine Unlearning Six-Way Evaluation for Language Models." The Thirteenth International Conference on Learning Representations.
>
> [2] Maini, Pratyush, et al. "Tofu: A task of fictitious unlearning for llms." arXiv preprint arXiv:2401.06121 (2024).

---

> ### Author Response · Authors · 2025-11-20
> **Response to Reviewer bZEQ (Part II)**
>
> **Q3. No experiments are provided to study how different step counts between the two optimization methods influence results. Moreover, there is no theoretical justification for using the same number of steps (N) for both optimizers. A rationale or ablation study would strengthen the methodological soundness.**
>
> **A3.** Thanks for this instructive feedback.
>
> We agree that optimizer scheduling in Hybrid optimization is an important factor. We respectfully clarify that our original submission already includes an ablation study examining the impact of different switching steps, as presented in Figure 7 and discussed in Lines 483-485. These results show that varying the switch steps N does not materially affect either unlearning performance or robustness, indicating that Hybrid FO-ZO remains stable across different step configurations.
>
> In response to the reviewer’s suggestion, we also conducted an additional study in which the two optimizers have different N. The results are summarized in **Table R4**.
> Our findings show that allocating an equal number of FO and ZO updates (Hybrid FO_20, ZO_20) achieves the best balance between unlearning effectiveness and robustness. This outcome is fully consistent with our method design grounded in the leader–follower game perspective (Sec. 5). When we assign more FO than ZO steps (e.g., Hybrid FO_20, ZO_10), robustness declines (shown by VerbMem on $D_f$ 72.1 with relearning). This is because the “leader” component (ZO updates responsible for steering the model toward robustness) becomes too weak relative to the “follower” FO updates that emphasize high-precision unlearning but do not explicitly promote robustness. Conversely, assigning more ZO than FO steps (e.g., Hybrid FO_10, ZO_20) slightly reduces unlearning effectiveness (shown by KnowMem on $D_r$ 52.0), since the “follower” (FO) becomes too weak to provide the high-fidelity updates needed to maintain strong unlearning performance.
>
> These results provide further empirical and conceptual justification for our choice of symmetric step counts (same N, like a zero-sum game) in Hybrid optimization and support the design rationale described in Lines 369-396 of the paper.
>
> **Table R4**: NPO-based unlearn performance and robustness with Hybrid optimization, evaluated on MUSE-Books.
>
> |  |VerbMem on $D_f$  | KnowMem on $D_f$  | KnowMem on $D_r$ | VerbMem on $D_f$ w. quantization | KnowMem on $D_f$ w. quantization | VerbMem on $D_f$ w. relearning | KnowMem on $D_f$ w. relearning |
> |-----------|-----------|-----------|----|------------------| ----| ---| ---|
> | Adam    |  0.0  |  0.0  | 53.6 | 48.5 | 34.8| 74.0 | 45.4 |
> | Hybrid (FO_20,ZO_20)    |   0.0  |   0.0  | 53.8 | 0.0 | 0.0 | 29.0 | 31.0 |
> | Hybrid (FO_10,ZO_20)    | 0.0    | 0.0    | 52.0 | 0.0 | 0.0 |  28.6 | 31.7 |
> | Hybrid (FO_20,ZO_10)    | 0.0 | 0.0   | 55.2 | 0.0 | 6.9 | 72.1 | 43.3 |
>
>
> **Q4. The proposed approach primarily combines two existing optimizers without introducing substantial conceptual innovation**
>
> **A4.** We thank the reviewer for the comment, though we respectfully disagree with the assessment that our approach lacks conceptual innovation. While our method indeed utilizes existing FO and ZO optimizers, the contribution of this work goes significantly beyond a simple combination of known components.
>
> First, we introduce optimizer grade as a new conceptual lens for understanding unlearning robustness, one that is agnostic to specific unlearning objectives. This perspective is both novel and central to our contributions. We show that optimizers differ not merely in algorithmic form but in the fidelity of descending information they employ (e.g., full-precision FO, gradient-compressed FO, ZO), and that this grade induces systematically different robustness behaviors. To our knowledge, this is the first work to formally articulate and empirically validate the link between optimizer grade and robustness in LLM unlearning.
>
> Second, our analysis reveals a previously unexplored connection between zeroth-order optimization and randomized smoothing. This provides a principled explanation for ZO’s robustness properties and is a key conceptual contribution of our paper, not previously discussed in unlearning literature.
>
> Third, our Hybrid FO–ZO framework is not a trivial mixture, but a methodologically grounded design motivated by a leader–follower game formulation (Sec. 5). In this view, the ZO component acts as the leader, steering the model toward robust, harder-to-disturb basins through structured noise, while the FO component acts as the follower, providing high-precision updates that preserve unlearning effectiveness. This perspective explains why alternate step-level hybridization yields benefits unattainable by either FO or ZO alone, and guides the choice of update ratios (validated empirically).

---

> ### Author Response · Authors · 2025-11-20
> **Response to Reviewer bZEQ (Part III)**
>
> **Q5. The robustness claim is insufficiently supported. To substantiate it, the method should be evaluated under membership inference and adversarial attack settings, which are standard in assessing unlearning and privacy resilience.**
>
> **A5.** We thank the reviewer for the insightful question. We respectfully clarify that our robustness claims concern **weight-space** perturbation robustness, not robustness against input-space or membership-oriented adversarial attacks. As stated in Lines 48–53 and 180–185, the scope of our work is to enhance the stability of unlearning under post-unlearning weight perturbations, a setting that is both practically important and conceptually connected to the optimizer used during unlearning.
>
> In particular, the perturbations we study, model quantization and adversarial weight tampering via relearning, explicitly target the model parameters. These forms of perturbation are directly influenced by the optimizer dynamics that shape the final unlearned weights. Our analysis introduces the novel concept of optimizer grade, showing that lower-grade optimizers (gradient-compressed FO and ZO) inherently bias optimization toward harder-to-disturb unlearning basins, thereby improving robustness against weight-space distortions. This weight-centric robustness and its connection to optimizer grade is, to our knowledge, new in the unlearning literature and agnostic to specific unlearning objectives. We believe that our work provides strong and sufficient evidence supporting the viewpoints and claims within the defined scope of our work.
>
> We agree that other robustness dimensions, such as membership inference or adversarial prompting, are valuable, yet they represent distinct attack modalities that are not the primary focus of this work. As is common in the unlearning and robustness literature, it is challenging for a single paper to comprehensively address every attack type. Yet, we greatly appreciate the reviewer’s suggestion and therefore conducted an additional experiment to assess whether optimizer downgrading offers incidental benefits against input-space jailbreak attacks. As shown in **Table R5**, downgraded optimizers (signAdam, ZO, and Hybrid) achieve lower Acc under enhanced GCG attacks compared with Adam, indicating improved resilience even beyond our primary threat model.
>
> **Table R5**: Acc on WMDP of different optimizers against jailbreak attacks, using the NPO-based unlearning objective.
>
> | Adam |  signAdam |ZO | Hybrid |
> |-----------|----| ---| ----|
> | 0.57 | 0.49 | 0.46 | 0.52 |

---

> > ### Author Response · Authors · 2025-11-24
> >
> > Dear Reviewer bZEQ,
> >
> > We hope this message finds you well.
> >
> > We are writing to kindly remind you that we have posted a detailed response to your constructive comments, and we have submitted the revised manuscript with the corresponding changes. We greatly appreciate the time you took to review our work, and we look forward to your engagement in the discussion.
> >
> > Thanks again for your time and effort.
> >
> > Best regards,
> >
> > The authors

---

> ### Author Response · Authors · 2025-11-26
>
> Dear Reviewer bZEQ,
>
> Following up on our recent submission of the revised paper and rebuttal responses, we kindly check if there are any questions or further clarifications needed.
>
> We genuinely hope our responses have adequately addressed your concerns. We remain fully available during the rest of the discussion period and welcome any additional feedback to ensure that the quality and technical depth of our work are clear.
>
> Thank you again for your time and engagement.
>
> Best regards,
>
> Authors

---

### Official Review · Reviewer_dqWt · 2025-11-01

**Soundness:** 2
**Presentation:** 2
**Contribution:** 2
**Rating:** 4
**Confidence:** 3

**Summary:**

The paper investigates how optimizers of different “grades”—namely 0th-order, 1st-order, and 2nd-order—affect the process of machine unlearning. The authors observe that lower-grade optimizers, while less effective at unlearning, are more robust to relearning. Building on this insight, they propose a hybrid optimizer that combines 0th- and 1st-order methods to achieve a balance between unlearning effectiveness and robustness. Experiments conducted on MUSE and WMDP support the findings.

**Strengths:**

* A timely and relevant exploration of the relationship between optimizer order and unlearning dynamics.
* The finding that lower-grade optimizers are more robust to relearning is intriguing and adds nuance to our understanding of unlearning mechanisms.
* The paper examines multiple optimizer variants, including SignedSGD, SignedAdam, and low-bit quantization, providing a richer empirical picture.

**Weaknesses:**

* While the observation that lower-grade optimizers are robust to relearning is interesting, the underlying cause may not be adequately analyzed. The discussion already hints that the robustness could stem from higher noise levels in these optimizers, which also naturally explain their weaker unlearning effects. If so, the “grade” of the optimizer might be a proxy variable rather than the true causal factor—making noise the key driver of robustness.
* The discussion of computational cost is largely absent. If noise is indeed the mechanism behind robustness, it could impose significant efficiency costs on the unlearning process. Consequently, the proposed hybrid approach may also entail higher cost. Since cost–performance trade-offs are crucial for practical deployment, this omission weakens the empirical contribution.
* The evaluation is limited to only three datasets, which feels insufficient for a paper with primarily empirical claims. Broader validation would help establish generality and robustness of conclusions.

**Questions:**

See Weaknesses, and also
1. Lines 177–180 appear to be repeated
1. Line 366: Should $k$ be even?

---

> ### Author Response · Authors · 2025-11-20
> **Response to Reviewer dqWt (Part I)**
>
> We thank the reviewer for the insightful comments. Please find our detailed responses to the identified weaknesses and questions below.
>
> **Q1. The observed robustness of lower-grade optimizers may actually result from their higher level noise, making noise, not optimizer grade, the true cause of robustness**
>
> **A1.** We thank the reviewer for raising this insightful question. We agree that downgraded optimizers, such as gradient-compressed methods (e.g., signSGD) and ZO optimizers, introduce higher stochasticity in their descending information compared with full-precision first-order optimizers.
>
> However, we would like to clarify that optimizer grade and noise should not be treated as two independent factors. As discussed in Lines 070–079, the grade of an optimizer is defined by the level and fidelity of the descending information it uses. For instance, gradient compression intentionally reduces information to lower bits, effectively injecting structured noise into the update direction. ZO optimization estimates gradients via noisy finite differences, producing an inherently lower-precision (higher-variance) update direction.
>
>
> **Thus, the introduction of noise is not an incidental side effect – it is precisely the mechanism that defines the optimizer grade.** When we refer to “optimizer grade,” we specifically mean the optimizer-induced noise structure that is inherently tied to how the optimizer processes descending information. This noise structure is systematic, tightly coupled to the optimizer’s update rule, and **importantly agnostic to the unlearning objective.** This explains why downgraded optimizers exhibit consistent behaviors across distinct unlearning objectives (e.g., NPO and GradDiff), as demonstrated in our experiments.
>
> In addition, we do not claim that “any form of noise” will lead to robustness. Instead, the robustness arises from the specific noise structure and optimization dynamics induced by downgraded optimizers: Gradient compression performs a natural denoising/projection (Eq. 3; Lines 268-269): multiple perturbed weight states are mapped to the same quantized representation, biasing optimization toward flatter, harder-to-disturb regions. ZO updates correspond to optimizing a randomized-smoothed objective (Eq. 5; Lines 303–308): the injected perturbations are not arbitrary but follow the finite-difference estimator that smooths the loss landscape in a principled manner. This creates a direct, established link between ZO optimization and optimization objective smoothing, not a generic “noise helps” effect. These mechanisms show that optimizer grade reflects structured, optimizer-driven perturbation, not mere randomness. This is further supported by the fact that downgraded optimizers such as signSGD and ZO methods have well-known convergence guarantees (e.g., [1–3]), meaning their behavior is far from uncontrolled stochasticity. Our empirical results reinforce this distinction: downgraded optimizers obtain similar pre-tampering unlearning performance to Adam (Figure 2(a–b), Figure 4(a–b)), contradicting the hypothesis that their effects are due to simple random noise injection, which would typically degrade unlearning effectiveness substantially.
>
> In summary, while noise is an inherent ingredient in downgraded optimizers, it is the optimizer-structured form of noise, captured by the notion of optimizer grade, that causally leads to robustness.
>
>
> [1] Bernstein, Jeremy, et al. "signSGD: Compressed optimisation for non-convex problems." International conference on machine learning. PMLR, 2018.
>
> [2] Liu, Sijia, et al. "A primer on zeroth-order optimization in signal processing and machine learning: Principals, recent advances, and applications." IEEE Signal Processing Magazine 37.5 (2020): 43-54.
>
> [3] Malladi, Sadhika, et al. "Fine-tuning language models with just forward passes." Advances in Neural Information Processing Systems 36 (2023): 53038-53075.

---

> ### Author Response · Authors · 2025-11-20
> **Response to Reviewer dqWt (Part II)**
>
> **Q2. The discussion of computational cost is absent. If noise is indeed the mechanism behind robustness, it could impose significant efficiency costs on the unlearning process.**
>
> **A2.** We thank the reviewer for raising this important point. As clarified in **A1**, the “noise” associated with downgraded optimizers is not arbitrary randomness but a structured, optimizer-induced perturbation that preserves convergence guarantees. Importantly, lower-grade optimizers do not inherently impose higher computational cost.
>
> First, consider zeroth-order (ZO) optimization: ZO avoids backpropagation and relies solely on forward function evaluations to estimate descent directions (Eq. (4)), making each iteration computationally cheaper. Although ZO may require more iterations to converge in full training scenarios, unlearning does not involve long-horizon optimization: its updates start from an already well-trained model and only need to make moderate adjustments. Consequently, the total number of ZO forward evaluations remains modest, leading to practically competitive wall-clock time overall (see empirical results in **Table R1**).
>
> Second, our additional empirical measurements (**Table R1**) show that downgraded optimizers such as ZO and our Hybrid FO–ZO method achieve comparable wall-clock time to standard first-order optimizers. This aligns with expectations: although ZO uses multiple function evaluations, the absence of backward passes and the limited number of unlearning iterations make its cost practically similar. We also observe that downgraded optimizers are significantly more efficient than the robust-optimization baseline SAM, whose sharpness-aware updates require an expensive inner loop. In contrast, our Hybrid method alternates between FO and ZO updates without introducing any nested optimization, preserving efficiency while improving robustness.
>
> **Table R1:** Run time (in minutes) comparison of GradDiff-based and NPO-based unlearning with different optimizers, evaluated on MUSE.
>
>
> MUSE-Books:
> |  | Adam | signSGD | signAdam | RS | SAM | ZO | Hybrid |
> |-----------|-----------|-----------|--- | ----| ----|--- | ---|
> | GradDiff   | 15.2    | 14.8    | 15.0 | 15.4 | 30.6 | 18.9 | 17.7 |
> | NPO    | 18.9 | 17.7   | 17.8 | 18.1 | 35.9 | 22.8 | 21.9 |
>
> MUSE-News:
> |  | Adam | signSGD | signAdam | RS | SAM | ZO | Hybrid |
> |-----------|-----------|-----------|--- | ----| ----|--- | ---|
> | GradDiff   | 33.2  | 31.1    | 32.6 | 35.9 | 85.7 | 40.8 | 41.7 |
> | NPO    | 39.3 | 38.5   | 39.0 | 39.8 | 81.3 | 40.9 | 39.3 |

---

> ### Author Response · Authors · 2025-11-20
> **Response to Reviewer dqWt (Part III)**
>
> **Q3.The evaluation is limited to only three datasets, which feels insufficient for a paper with primarily empirical claims.**
>
> **A3.** We respectfully disagree with the reviewer’s assessment. Our study has evaluated on datasets spanning three major and widely adopted unlearning scenarios. These include (1) data memorization unlearning: MUSE-Books and MUSE-News (Figures 1–4); (2) harmful or unsafe knowledge unlearning: WMDP-Bio (Figure 5); and (3) fictitious concept/entity unlearning: TOFU (Figure 6). These benchmarks cover most of the representative datasets used in the current LLM unlearning literature. As documented in prior works [4–6], MUSE, WMDP, and TOFU are considered the standard, diverse, and complementary testbeds for evaluating unlearning across memorization, safety-critical knowledge removal, and structured fact deletion. Many prior papers evaluate on an equal or smaller subset of these datasets. In addition, our experiments span multiple unlearning objectives and multiple robustness stressors (weight quantization and relearning), providing broad validation across different datasets. Therefore, although no evaluation suite can be exhaustive, we believe our study already covers most of the representative and widely accepted benchmarks and sufficiently supports the empirical generality of our conclusions.
>
> Last, we would like to respectfully clarify that characterizing our work as “primarily empirical” may overlook several of our conceptual and analytical contributions. Beyond extensive experiments, our paper introduces the *optimizer grade* as a principled lens for understanding unlearning robustness; provides the rationale for the robustness gains of ZO optimization via its connection to randomized smoothing (Sec. 4); and proposes the FO–ZO hybrid through a leader–follower game perspective that explains why alternating optimization naturally couples unlearning effectiveness and robustness (Sec. 5). These components collectively provide conceptual insights that go beyond empirical observation and advance the technical understanding of robust LLM unlearning.
>
>
>
> **Q4. Lines 177-180 appears to be repeated.**
>
> **A4.** We thank the reviewer for this important finding. We will correct this typo in our revision.
>
> **Q5. Should k be even?**
>
> **A5.** We thank the reviewer for this comment and we apologize for this typo. k should indeed be even since Hybrid optimization starts with FO and ends with ZO.
>
> [4] Fan, Chongyu, et al. "Towards LLM Unlearning Resilient to Relearning Attacks: A Sharpness-Aware Minimization Perspective and Beyond." Forty-second International Conference on Machine Learning.
>
> [5] Wang, Changsheng, et al. "Invariance Makes LLM Unlearning Resilient Even to Unanticipated Downstream Fine-Tuning." Forty-second International Conference on Machine Learning.
>
> [6] Wang, Yue, et al. "GRU: Mitigating the Trade-off between Unlearning and Retention for LLMs." Forty-second International Conference on Machine Learning.

---

> > ### Author Response · Authors · 2025-11-24
> >
> > Dear Reviewer dqWt,
> >
> > We hope this message finds you well.
> >
> > We are writing to kindly remind you that we have posted a detailed response to your constructive comments, and we have submitted the revised manuscript with the corresponding changes. We greatly appreciate the time you took to review our work, and we look forward to your engagement in the discussion.
> >
> > Thanks again for your time and effort.
> >
> > Best regards,
> >
> > The authors

---

> > > ### Comment · Reviewer_dqWt · 2025-11-25
> > >
> > > Thank you for the detailed response and additional empirical results. I found the clarification especially helpful that it is not arbitrary noise, but the _structured_ noise induced by lower-grade optimizers, that drives the effect. Overall, the rebuttal addresses most of my concerns. I will increase my score to 6.

---

> > > > ### Author Response · Authors · 2025-11-25
> > > >
> > > > Dear Reviewer dqWt,
> > > >
> > > > Thank you again for your thoughtful comments and for taking the time to reconsider our response. We are glad that our clarifications addressed your concerns, and we truly appreciate your revised score and encouragement.
> > > >
> > > > Thank you again for your help and support!
> > > >
> > > > Authors,

---

### Author Response · Authors · 2025-11-29
**Summary of response to all reviewers (Part I)**

Dear ACs, SACs, and PCs,

We sincerely appreciate the constructive feedback provided by the reviewers and are grateful for their active engagement throughout the rebuttal period. In **our rebuttal (submitted on Nov. 19, 2025)** and the subsequent **revised manuscript (submitted on Nov. 23, 2025)**, we made substantial efforts to clarify all raised concerns, address technical questions, and enrich experimental results.

Before the system bug spread and affected the displayed reviewer scores, our rebuttal had already led to clear positive updates from multiple reviewers: **Reviewer dqWt** increased their rating to 6 (updated on Nov. 24, 2025); **Reviewer dWKo** increased their rating to 8 (updated on Nov. 22, 2025); **Reviewer psAf** confirmed and maintained their positive score of 6 (responded on Nov. 22, 2025). Only **Reviewer bZEQ** did not provide any follow-up comments nor acknowledge our responses.

As a result, **the true updated score profile before the system reverted the ratings was 8, 6, 6, 4** (all provided before Nov. 24), reflecting reviewers’ improved assessments following our response compared to the reverted pre-discussion scores of 6, 4, 4, 4.

Given this context, we provide below a concise **summary of our rebuttal clarifications** and the reviewers’ updated feedback during the discussion period.


**Summary of response to Reviewer dqWt (score from 4 to 6, updated on Nov. 24, 2025)**
1. We clarified that optimizer grade, not random noise, is the fundamental factor driving robustness, and provided additional explanations to resolve this misunderstanding (see Response A1).
2. We added runtime and efficiency evaluations, showing that downgraded optimizers not only enhance robustness but also improve computational efficiency (see Response A2).
3. We clarified why our experimental validations are sufficient and appropriate for the scope of the paper, addressing the reviewer’s concerns on completeness (see Response A3).
4. We corrected identified typos and further improved the  presentation quality in the revised manuscript.

Post-rebuttal, Reviewer dqWt positively commented:
> I found the clarification especially helpful that it is not arbitrary noise, but the structured noise induced by lower-grade optimizers, that drives the effect. Overall, the rebuttal addresses most of my concerns. I will increase my score to 6.

**Summary of response to Reviewer bZEQ (score 4, no follow-up)**
1. We clarified that the reviewer-suggested evaluations were already included in our original submission, and pointed the reviewer to the relevant sections (see Response A1).
2. We provided additional experiments on the convergence of our proposed method (see Response A2).
3. We clarified the ablation study on Hybrid optimization, and provided additional experiments to further validate its design rationale (see Response A3).
4. We further clarified the theoretical background and design novelty of Hybrid optimization (see Response A4).
5. We clarified that our paper focuses specifically on weight-space robustness, and we do not claim robustness to input-space adversarial attacks. But to provide additional context, we also included robustness evaluations against jailbreak attacks as a complementary analysis (see Response A5).

Despite our substantial rebuttal efforts, Reviewer bZEQ did not engage in the discussion phase.

---

> ### Author Response · Authors · 2025-11-29
> **Summary of response to all reviewers (Part II)**
>
> **Summary of response to Reviewer dWKo (score from 4 to 8, updated on Nov. 22, 2025)**
> 1. We clarified that optimizer downgrade does not conflict with IRM and SAM (see Response A1).
> 2. We clarified why we use Adam instead of SGD as the first-order optimizer (see Response A2).
> 3. We clarified that our manuscript already included the evaluation of the second-order optimizer (see Response A3).
> 4. We clarified that retention performance is consistently evaluation in our paper, and provided calibrated unlearning evaluation via model merging (see Response A4).
> 5. We provided rationale for our hyperparameter setting (see Response A5).
> 6. We further clarified how we establish the claim that optimizer downgrade leads to robustness upgrade (see Response A6).
> 7. We included the additional related works (see Response A7).
>
> Post-rebuttal, Reviewer dWKo positively commented:
> >The authors' feedback is decent and clear, which addresses much of my concerns. Accordingly, I increase my score to 8.
>
> **Summary of response to Reviewer psAf (score from 6 to 6, responded on Nov. 22, 2025)**
>
> 1. We further clarified why optimizer downgrade leads to unlearning robustness (see Response A1).
> 2. We clarified why our work specifically focuses on LLM unlearning (see Response A2).
> 3. We provided additional experiments on runtime evaluation (see Response A3).
> 4. We further clarified the design rationale of Hybrid optimization from a leader–follower game perspective and provided additional experiments to justify and strengthen this design choice (see Response A4).
>
> Post-rebuttal, Reviewer psAf appreciated our clarifications and commented:
> >I have read the authors’ response carefully and appreciate the additional clarifications. Although I am not an expert in unlearning, the paper remains solid in its contributions and technical quality. I therefore maintain my positive score.
>
> Thank you once again for your time, effort, and thoughtful consideration in reviewing our submission.
>
> Authors

---

### Meta-Review · Area_Chair_hDhR · 2026-01-08

**Summary:**

This paper investigates the role of optimizer "grade" (the level of information it exploits) in shaping unlearning robustness. The authors demonstrate that "downgrading" the optimizer, e.g., using zero-order (gradient-free) or compressed-gradient methods instead of standard first-order ones like Adam, could lead to unlearning that is significantly more robust. They also propose a hybrid optimizer that combines first-order and zero-order updates to preserve unlearning efficiency while enhancing robustness. The reviewers found the work intriguing and providing a fresh insight, and praised the strong empirical evidence and the effectiveness of the hybrid optimizer. They also raised several concerns on the true causal factor for robustness (noise or optimizer grade), the lack of computational cost evaluation, and some of the limitations in evaluations.

**Reviewer Concerns:**

Reviewers dqWt, dWKo, and psAf responded to the rebuttal and stated that the rebuttal addressed much of their concerns. Reviewer bZEQ did not respond, although most of their concerns seem to be convincingly addressed by the rebuttal.

**Reviewer Scores:**

Reviewers dqWt, dWKo, and psAf have either increased or maintained their cores, from 4, 4, 6 to 6, 8, 6. While reviewer bZEQ did not respond, it is more likely that they would increase their score given the detailed and convincing rebuttal.

---

### Decision · Program_Chairs · 2026-01-26

Accept (Poster)